# Locomotor recovery following contusive spinal cord injury does not require oligodendrocyte remyelination

Greg J. Duncan[1,2], Sohrab B. Manesh[1,3], Brett J. Hilton[1,2,6], Peggy Assinck [1,3], Jie Liu[1], Aaron Moulson[1,2], Jason R. Plemel [4] & Wolfram Tetzlaff[1,2,5]

Remyelination occurs after spinal cord injury (SCI) but its functional relevance is unclear. We assessed the necessity of myelin regulatory factor (*Myrf*) in remyelination after contusive SCI by deleting the gene from platelet-derived growth factor receptor alpha positive (PDGFRα-positive) oligodendrocyte progenitor cells (OPCs) in mice prior to SCI. While OPC proliferation and density are not altered by *Myrf* inducible knockout after SCI, the accumulation of new oligodendrocytes is largely prevented. This greatly inhibits myelin regeneration, resulting in a 44% reduction in myelinated axons at the lesion epicenter. However, spontaneous locomotor recovery after SCI is not altered by remyelination failure. In controls with functional MYRF, locomotor recovery precedes the onset of most oligodendrocyte myelin regeneration. Collectively, these data demonstrate that MYRF expression in PDGFRα-positive cell derived oligodendrocytes is indispensable for myelin regeneration following contusive SCI but that oligodendrocyte remyelination is not required for spontaneous recovery of stepping.

[1] International Collaboration on Repair Discoveries (ICORD), University of British Columbia (UBC), 818 West 10th Avenue, V5Z 1M9 Vancouver, BC, Canada. [2] Department of Zoology, University of British Columbia, 4200-6270 University Blvd, Vancouver V6T 1Z4 BC, Canada. [3] Graduate Program in Neuroscience, University of British Columbia, 3402-2215 Wesbrook Mall, Vancouver V6T 1Z3 BC, Canada. [4] The Department of Clinical Neurosciences, Hotchkiss Brain Institute, University of Calgary, 3330 Hospital Drive NW, T2N 4N1 Calgary, AB, Canada. [5] Department of Surgery, University of British Columbia, 2775 Laurel Street, Vancouver V5Z 1M9 BC, Canada. [6]Present address: Deutsches Zentrum für Neurodegenerative Erkrankungen (DZNE), Sigmund-Freud-Straße 27, 53127 Bonn, Germany. These authors contributed equally: Greg J. Duncan, Sohrab B. Manesh.  Correspondence and requests for materials should be addressed to W.T. (email: tetzlaff@icord.org)

Spinal cord injury (SCI) can lead to severe and permanent motor, sensory, and autonomic dysfunction due to the adult mammalian spinal cord's inability to regenerate lost neurons and their connections[1]. Most SCIs in humans do not result in the complete transection of the spinal cord, but instead axons are spared at the lesion epicenter[2], and a period of limited functional improvement commences soon after SCI despite axon regeneration failure[3,4]. Enhancing the functional connectivity of the spared circuitry may be a viable means of promoting functional improvements following SCI[5]. However, oligodendrocyte death in the weeks after SCI[6] presumably results in the demyelination of spared axons[7–10], which could diminish the functionality of spared circuits. Demyelination impairs the amplitude and speed of electrical conductance[11–13] and oligodendrocyte loss may leave axons vulnerable to degeneration[14,15]. For these reasons, strategies to enhance oligodendrocyte remyelination of spared axons have been hypothesized to promote functional improvements following SCI[16–19].

Myelin regeneration is a spontaneous process: new oligodendrocytes and Schwann cells regenerate lost myelin in the absence of therapeutic intervention[20–27]. Platelet-derived growth factor receptor alpha (PDGFRα) expression in resident, nonvascular associated cells, identifies these cells as oligodendrocyte progenitor cells (OPCs)[28,29], which differentiate into new oligodendrocytes after SCI[20,22,26]. Ependymal cells can also contribute to oligodendrocyte production, albeit minimally[20,30]. Intricate transcriptional regulation is required for OPCs to differentiate into new myelinating oligodendrocytes. During both development and myelin regeneration, the transcription factor myelin regulatory factor (Myrf) is essential for OPC differentiation and myelin protein expression[31,32]. Nevertheless, the role of MYRF has not been elucidated after SCI, nor whether PDGFRα + OPCs constitute an indispensable source of remyelinating oligodendrocytes.

The functional relevance of oligodendrocyte remyelination after SCI is also unclear. Remyelination in the spinal cord is correlated with improvements in locomotion following chemical demyelination and after the consumption of an irradiated diet[33,34]. Transplantation of cells capable of forming new oligodendrocytes after SCI is coupled with functional improvements when the overall extent of remyelination is increased[35–38]. Endogenous myelin regeneration is an efficient process after SCI, as indicated by the presence of numerous thinly myelinated axons[10,13,39], shorter internodes[9,24,40], and by fluorescently labeling new myelin in transgenic mice[22,26]. Given the extent of endogenous oligodendrocyte remyelination, it is plausible that remyelination contributes to the limited level of locomotor recovery after SCI. However, axons are capable of conducting through short segments of demyelination in vivo[41], and the extent of demyelination among intact axons may not be sufficient to contribute to detectable functional decline. Despite this, myelin regeneration is the mechanistic basis of several ongoing clinical trials and has become an important[16,19], yet contentious[17,18,42] therapeutic target.

To ascertain the role of oligodendrocyte myelin regeneration in locomotor recovery, we used transgenic mice, which permit the selective ablation of oligodendrocyte remyelination. Oligodendrocyte remyelination requires the differentiation of OPCs into new oligodendrocytes[43,44], so we deleted Myrf, crucial for OPC differentiation[31,32], prior to moderate thoracic spinal cord contusion injury in mice. We find that MYRF is essential for both the accumulation of new oligodendrocytes and for remyelination. Schwann cell myelination is not altered by Myrf deletion from PDGFRα + cells, nor does the extent of Schwann cell myelination increase to compensate for a failure of oligodendrocyte remyelination. This demonstrates that effective remyelination requires local PDGFRα + progenitors to differentiate and express MYRF after SCI to generate new oligodendrocytes. Surprisingly, the recovery of hindlimb motor function assessed on open field testing, the horizontal ladder and Catwalk gait analysis is unaltered by the deletion of Myrf from OPCs. Further, by labeling new myelin, we demonstrate that nearly all new oligodendrocyte myelin forms after the initial recovery of hindlimb stepping in mice with functional MYRF. These data indicate that while spontaneous oligodendrocyte remyelination is extensive following SCI, it is not associated with improvements in hindlimb motor function during spontaneous recovery in this model.

## Results

**Effective recombination in OPCs after SCI in Myrf ICKO mice.** The cellular mechanisms that drive locomotor improvements following SCI are poorly understood. Genetic fate mapping reveals extensive remyelination by resident OPCs differentiating into new oligodendrocytes in response to SCI[26], however, the extent to which oligodendrocyte remyelination contributes to spontaneous motor improvements is unknown. Removing a gene, like Myrf, essential for OPC differentiation should halt remyelination in response to SCI[31]. This would enable an assessment of the role of endogenous oligodendrocyte remyelination in functional improvements. We crossed mice carrying LoxP sites flanking both copies of exon 8 (homozygous) of the Myrf gene (Myrf^fl/fl) with mice expressing a tamoxifen-inducible Cre recombinase under the PDGFRα reporter to produce Myrf^fl/fl PDGFRα-CreERT2 mice (Myrf ICKO). When tamoxifen is administered, recombination occurs in PDGFRα + OPCs, resulting in excision of exon 8 of the Myrf gene in Myrf ICKO mice (Fig. 1a), thereby rendering this critical transcription factor nonfunctional[31,32,45]. Control mice were littermate Myrf^fl/fl mice which lacked PDGFRα-CreERT2, so when tamoxifen is administered exon 8 is not removed and the gene remains functional (Fig. 1a). Adult mice were pretrained on behavioral tasks then dosed with tamoxifen prior to injury (Fig. 1b). Mice were injured with a moderate contusive injury known to induce demyelination of spared axons[9,13,27]. Like most human injuries, moderate contusions have axon sparing and demonstrate limited locomotor improvement. There were no differences in injury force or displacement applied by the infinite horizon (IH) impactor between Myrf ICKO and controls (Fig. 1c, d).

We examined the effectivess of tamoxifen to induce recombination in the spinal cord of both Myrf^wt/wt and Myrf^fl/fl mice heterozygous for the PDGFRα-CreERT2 and the Rosa26 mGFP (mT/mG) transgenes after injury. These mice expressed membrane tethered fluorescence in response to Cre-mediated recombination, permitting morphological and phenotypical assessment of these recombined cells (Fig. 1e). Recombination within OPCs (defined as OLIG2+ PDGFRα+ double-positive cells) at 6 weeks postinjury (WPI) resulted in mGFP expression in $88.3 \pm$ (standard error of the mean) 2.9% of control and $89.4 \pm$ 1.6% of Myrf ICKO mice OPCs (Fig. 1g). Thus, recombination was highly effective and labeled the majority of OPCs. As OPCs differentiate during remyelination they begin to express MYRF[31]. Accordingly, we found that MYRF was expressed only in CC1 + OLIG2 + oligodendrocytes and was not expressed in oligodendrocyte lineage cells which had not differentiated (OLIG2+ CC1-negative) (Fig. 1h). EdU (5-ethynyl-2′-deoxyuridine) was administered after SCI and effectively labels proliferative cells including OPCs and can be used to distinguish newly differentiated oligodendrocytes[46]. In Myrf ICKO mice, MYRF was nearly absent from new oligodendrocytes labeled with CC1+ and EdU, in contrast to controls where EdU+ nuclei was observed in MYRF+ CC1+ cells (Fig. 1i). Myrf deletion from OPCs did not alter the extent of spared tissue indicated by glial

fibrillary acidic protein (GFAP) staining (Fig. 1j) at any point examined between 800 μm rostral to caudal of lesion epicenter following thoracic SCI (Fig. 1k). Collectively, these data demonstrated that *Myrf* ICKO mice can be used to effectively induce recombination in OPCs, thereby reducing MYRF

expression in new oligodendrocytes in response to SCI, but did not alter injury dynamics or tissue sparing.

**Myrf is required for oligodendrocyte accumulation after SCI.** We next determined if *Myrf* ICKO was effective at inhibiting the

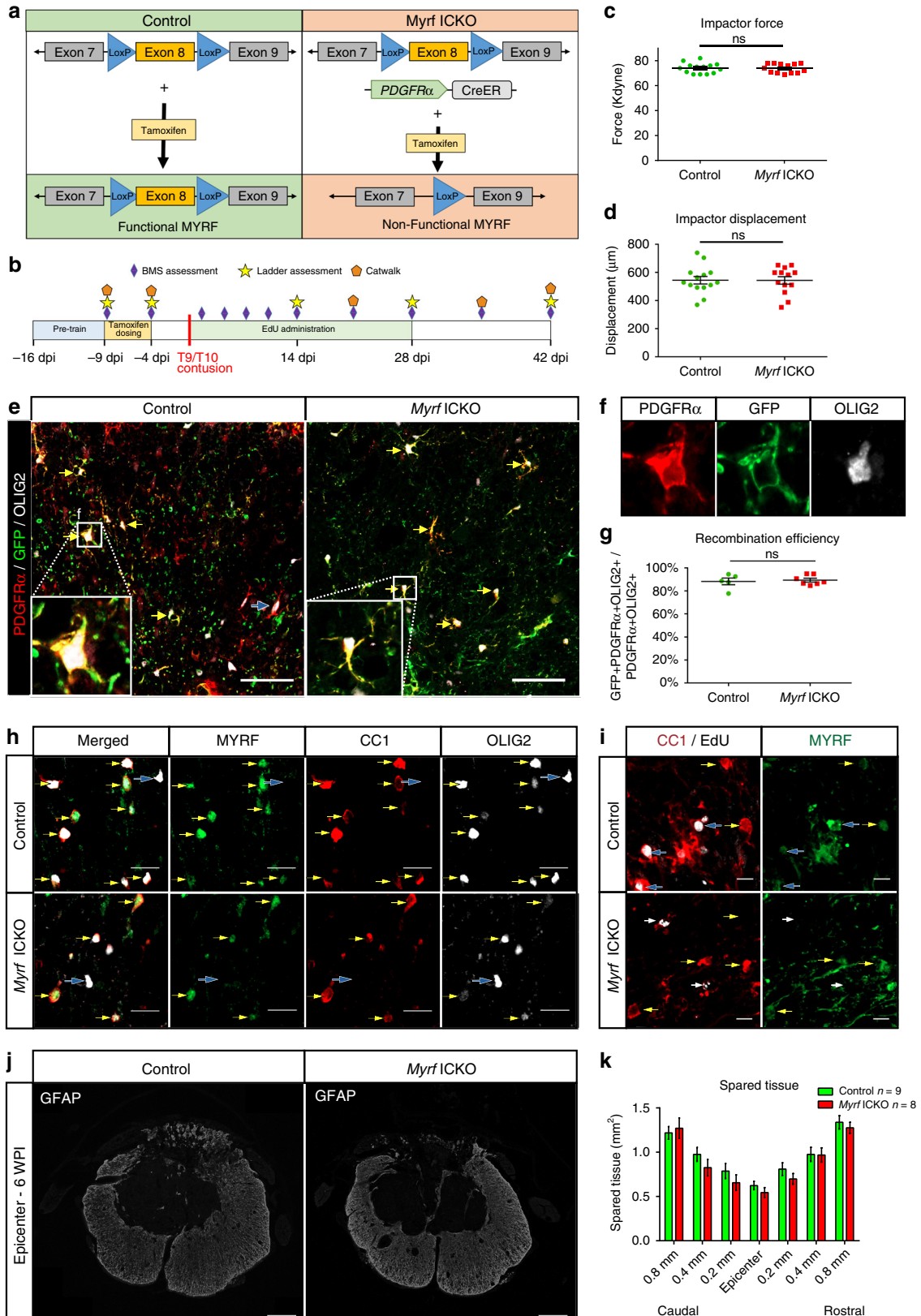

accumulation of new oligodendrocytes in response to SCI. Oligodendrocyte lineage cells (OLIG2+) were typically not found within the lesion, but present in the spared white matter (Fig. 2a, b). To identify new oligodendrocytes, we examined EdU expression in OLIG2+ CC1+ cells. (Fig. 2c, d). There was over an 11-fold reduction in new oligodendrocytes (EdU+ CC1+ OLIG2+) in *Myrf* ICKO mice compared to the controls (1286 ± 250 to 14,315 ± 1308 mm³), indicating that *Myrf* ICKO almost completely prevented the accumulation of new oligodendrocytes following SCI (Fig. 2e). *Myrf* ICKO mice also had fewer total oligodendrocytes (CC1+ OLIG2+) than control mice, with a mean of 14,760 ± 2233 cells mm⁻³ in *Myrf* ICKO mice compared to 34,034 ± 2585 cells mm⁻³ in control mice (Fig. 2e). Additionally, the mean difference in total oligodendrocyte densities (19,274 cells mm⁻³) was mostly accounted for by the lack of new oligodendrogenesis in *Myrf* ICKO, as nearly this many EdU+ oligodendrocytes (14,315 ± 1308 cells mm⁻³) were produced in WT mice after SCI. *Myrf* ICKO was successful at preventing the accumulation of new oligodendrocytes at all distances examined from lesion epicenter (Fig. 2f). OPC density (PDGFRα+ OLIG2+) did not differ between *Myrf* ICKO and controls, nor did the density of OPCs which have proliferated (EdU+ PDGFRα+ OLIG2+) (Fig. 2g). Therefore, MYRF is not required for the proliferation or recruitment of OPCs after SCI but inhibits the accruement of new oligodendrocytes.

***Myrf* ICKO prevents oligodendrocyte remyelination.** We next determined whether *Myrf* knockout from resident OPCs was sufficient to halt new myelination in response to SCI. Six weeks after a moderate thoracic contusion injury, there was sparing of some ventrolateral white matter at the lesion epicenter (Fig. 1j, k), which contains both undamaged and regenerated myelin. To unequivocally differentiate newly generated myelin from surviving myelin, we crossed *Myrf* ICKO mice with Rosa26mGFP (mT/mG) mice. When administered tamoxifen, OPCs with active Cre recombinase express a membrane-anchored GFP that can be visualized within new myelin produced by oligodendrocytes which have differentiated from OPCs (Fig. 3a, b)[24,26,31,44]. By six WPI, in the ventrolateral white matter, control mice had new myelin sheaths, which were indicated by GFP + colabeling within MBP + sheaths around NF-200/SMI312 positive axons (Fig. 3c, e). Conversely, in *Myrf* ICKO mT/mG mice, GFP processes wrapped axons, but were almost always negative for MBP (Fig. 3d, f). After 6 weeks, the *Myrf* ICKO mice had generated only 248 ± 56 new myelin sheaths mm⁻² in contrast to control

mice which had 4664 ± 674 sheaths mm⁻² (Fig. 3g). Overall, 1.7 ± 0.4% of the myelinated axons at the lesion epicenter in *Myrf* ICKO had new myelin as compared to 28.4 ± 3.0% in control mice (Fig. 3h). There was also a small population of axons that were wrapped by GFP + processes, but did not express MBP in both control and *Myrf* ICKO mice and this may represent an early stage of axon ensheathment by oligodendrocyte lineage cells (Fig. 3d, f) and did not differ between groups (Fig. 3i). At two WPI, there were very few GFP + processes that colabeled with MBP + myelin sheaths in both control (Fig. 3j) and *Myrf* ICKO mice (Fig. 3k) indicating little remyelination occurred in the first two WPI (Fig. 3g). In control mice, there is a higher density of myelin sheaths at six WPI when compared to two WPI animals ($F_{(1, 18)} = 55.07$ two-way repeated measures ANOVA, $P < 0.001$; two WPI vs. six WPI: $P < 0.001$ Tukey's post hoc test) demonstrating considerable myelinogenesis during that time. Taken together, endogenous oligodendrocyte remyelination after SCI occurs largely after two WPI and this was almost completely prevented by removing *Myrf* from OPCs.

**Schwann cell myelination is unaltered by *Myrf* ICKO after SCI.** The majority of new (GFP+) myelin sheaths in *Myrf* ICKO mice were found in the dorsal column, a location of extensive Schwann cell myelination following dorsal SCI[26,27]. Given that MBP is not only expressed in oligodendrocyte myelin but also Schwann cell myelin[47], we determined whether the new myelin produced in *Myrf* ICKO mice were derived from Schwann cells. The myelin protein zero (P0) is a Schwann cell-specific myelin marker and can be used to distinguish Schwann cell myelin from oligodendrocyte myelin[26,27]. At both two and six WPI, axons were wrapped by P0+ myelin in both controls and *Myrf* ICKO mice, some of which was produced by recombined cells (mGFP+) (Fig. 4a). The presence of GFP+ P0+ myelin supports our previous findings that PDGFRα+ cells produce Schwann cells after SCI[26]. Using higher magnification confocal microscopy, we find clear colabeling of P0 with GFP in the dorsal column in control (Fig. 4b) and *Myrf* ICKO mice (Fig. 4d), but only rare P0 + sheaths in the ventralolateral white matter of the spinal cord in either group (Fig. 4c, e). *Myrf* ICKO mice had no difference in the density of P0+ Schwann cell myelin relative to controls at two or six WPI (Fig. 4f). Similarly, the density of PDGFRα-derived Schwann cell myelin (GFP+ P0+) was not different in *Myrf* ICKO mice relative to control mice at two or six WPI (Fig. 4g), nor was the percentage of P0+ myelinated axons derived from PDGFRα+ cells (two WPI: 17.5% ± 4.0% for controls,

---

**Fig. 1** *Myrf* ICKO mice have effective recombination in OPCs following moderate thoracic SCI. **a** Illustration of transgenes used in this experiment. *Myrf* ICKO mice were generated by crossing mice with exon 8 of *Myrf* floxed with mice with the PDGFRα-CreERT2 transgene to produce *Myrf*^fl/fl PDGFRα-CreERT2 mice. Control mice lacked the PDGFRα-CreERT2 transgene. **b** Illustration of experimental timeline. **c** Impact force (kilodynes) imparted on the spinal cord during SCI indicates no difference between groups ($df = 25$, $t = 0.103$, $P = 0.912$, Student's $t$ test). **d** Displacement (μm) of the impactor tip upon contact with the spinal cord during thoracic contusion shows no statistical difference between groups ($df = 25$, $t = 0.037$, $P = 0.971$, Student's $t$ test). **e** Overview images from the ventrolateral white matter adjacent to the lesion epicenter in control and *Myrf* ICKO mice crossed with a tamoxifen inducible reporter that tethers GFP to the membrane (mT/mG). The majority of PDGFRα + OLIG2 + cells are recombined (GFP expression, yellow arrows), but occasional nonrecombined PDGFRα + cells are observed (PDGFRα + GFP−, blue arrows). **f** Inlays of single optical sections demonstrating colabeling of PDGFRα with GFP in OLIG2 + cells. **g** Quantification of the recombination efficiency in OPCs at six WPI. There is no difference in recombination between control and *Myrf* ICKO mice ($df = 10$, $t = 0.368$, $P = 0.627$, Student's $t$ test). **h** Single optical confocal section micrographs demonstrating colabeling of MYRF in CC1 + OLIG2 + oligodendrocytes (yellow arrows). OLIG2 + cells lacking CC1 do not have MYRF expression in either group (blue arrows). **i** Single optical section from control or *Myrf* ICKO mice demonstrating colabeling of MYRF in CC1 + EdU + oligodendrocytes (blue arrows) in control mice, but not in CC1 + EdU + oligodendrocytes in *Myrf* ICKO mice (white arrows). **j** Spinal cord cross-section of the lesion epicenter stained for GFAP at six weeks post injury (WPI) in *Myrf* ICKO and control mice. **k** Quantification of GFAP + spared tissue at different distances from lesion epicenter. There is no significant difference between *Myrf* ICKO and control mice at any given distance from lesion epicenter (multiple Student's $t$ test with Holm-Šídák correction, epicenter $t = 1.095$, $P = 0.291$) ns non-significant. Scalebars = 50 μm (**e**), 10 μm (**h**), 5 μm (**i**), 100 μm (**j**) . Error bars are mean ± SEM

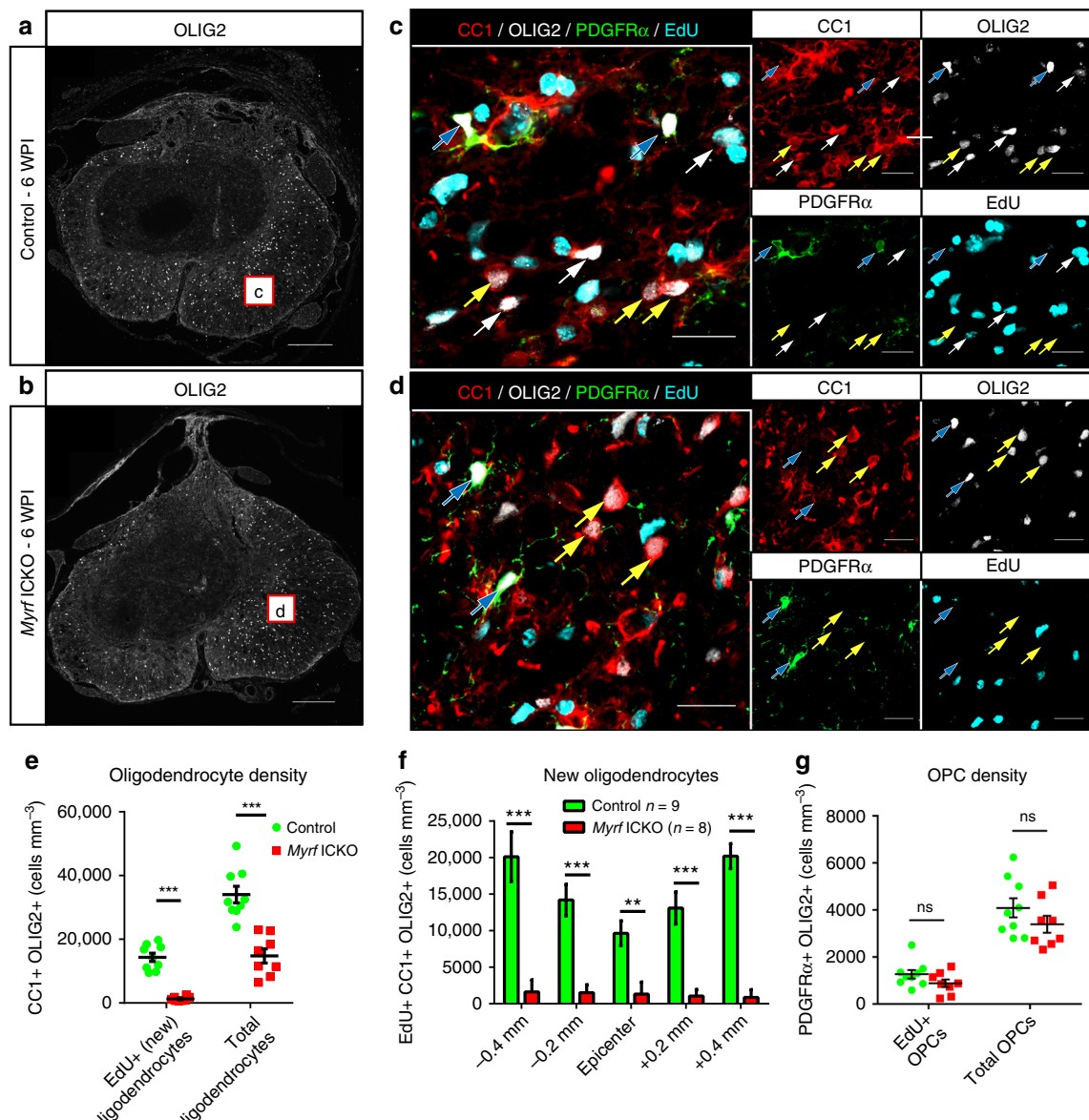

**Fig. 2** *Myrf* ICKO mice generate few new oligodendrocytes in response to SCI. Overview of OLIG2 staining at injury epicenter in **a** control and **b** *Myrf* ICKO mice at six WPI. Boxes are approximate areas where **c** and **d** were imaged. **c**, **d** Example high magnification representative images of the ventrolateral white matter stained with PDGFRα, OLIG2, CC1, and EdU in control and *Myrf* ICKO mice. Single channel images are displayed separately on the right. Yellow arrows indicate oligodendrocytes lacking EdU (OLIG2+ CC1+ EdU-negative), which are likely spared oligodendrocytes, while white arrows indicate new oligodendrocytes (OLIG2+ CC1+ EdU+) and blue arrows indicate OPCs (OLIG2+ PDGFRα+ EdU±). There are very few new oligodendrocytes following SCI in *Myrf* ICKO. **e** Quantification demonstrates control mice have a higher density of new oligodendrocytes (CC1+ OLIG2+ EdU+) ($df = 15$, $t = 9.224$, $P < 0.001$, Student's $t$ test) and total oligodendrocytes (CC1+ OLIG2+) ($df = 15$, $t = 5.570$, $P < 0.001$, Student's $t$ test) compared to *Myrf* ICKO animals. **f** Distribution of newly generated cells at different distances from lesion epicenter (OLIG2+ CC1+ EdU+). At all distances, control mice have more new oligodendrocytes relative to *Myrf* ICKO mice (multiple Student's $t$ test with Holm-Šídák correction, epicenter $t = 4.100$, $P = 0.001$, all others distances $P < 0.001$). **g** Quantification of the density of OPCs that have proliferated (PDGFRα+ OLIG2+ EdU+) and total density of OPCs (PDGFRα+ Olig2+) indicate there is no statistical difference between *Myrf* ICKO and controls (total OPC density: $df = 15$, $t = 1.535$, $P = 0.146$; proliferative OPC density: $df = 15$, $t = 1.267$, $P = 0.225$, Student's $t$ tests). **P ≤ 0.01 ***P ≤ 0.001. Scale bars = 100 μm (**a**, **b**), 20 μm (**c**, **d**). Error bars are mean ± SEM

13.6% ± 5.1% for *Myrf* ICKO at six WPI: 23.8% ± 9.9% for controls, 20.5% ± 3.0% for *Myrf* ICKO) (Fig. 4h), demonstrating that MYRF was not required for Schwann cell myelination from PDGFRα+ cells. While we previously found the majority of Schwann cells were PDGFRα+ cell derived, the percentage of PDGFRα-derived Schwann cell myelin increases over time and the quantification in this study was undertaken at an earlier time point than previous studies[26]. In *Myrf* ICKO mice by six WPI, the

total density of myelin produced by recombined cells (GFP+ MBP+, 248 ± 56 sheaths mm$^{-2}$) (Fig. 3d) could be almost entirely accounted for by the amount of Schwann cell myelination (GFP+ P0+, 240 ± 39 myelin sheaths mm$^{-2}$) (Fig. 4f). Therefore, MYRF is dispensable for Schwann cell myelination in the central nervous system (CNS) after SCI and impairing oligodendrocyte remyelination does not cause a compensatory increase in Schwann cell myelination.

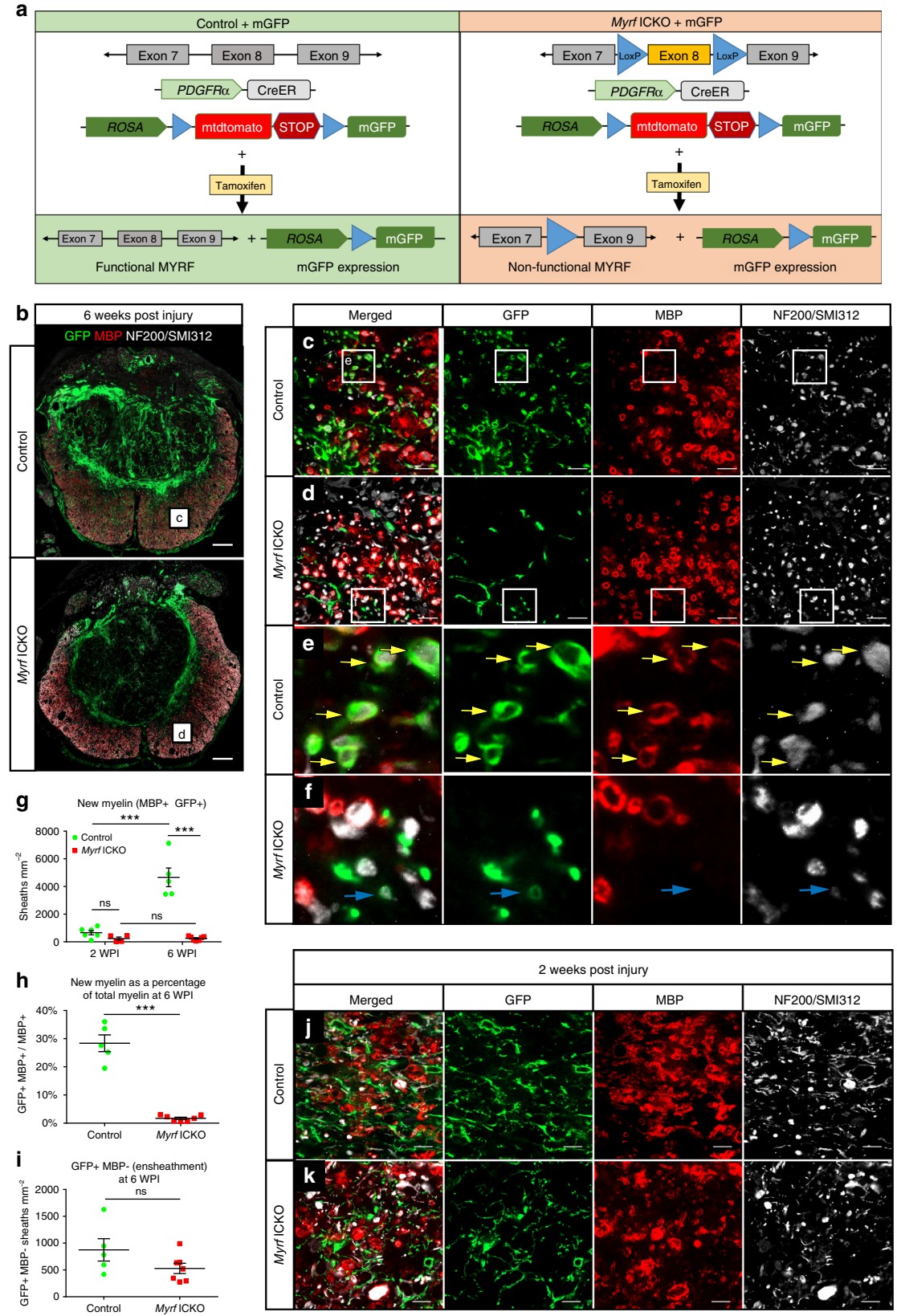

**Myrf ICKO results in chronic demyelination after SCI.** Typically, remyelination is efficient in rodents with little evidence of chronic demyelination after SCI[9,40]. Genetic fate mapping revealed that recombined cells in *Myrf* ICKO mice are nearly unable to produce new oligodendrocyte myelin following SCI, suggesting persistent demyelination may be present. However, de novo myelination does not require overt demyelination[46], nor does it reveal differences in the total level of myelination between groups. We visualized resin-embedded sections at the lesion epicenter of *Myrf* ICKO and control mice to determine the degree

**Fig. 3** *Myrf* ICKO blocks nearly all oligodendrocyte remyelination in recombined cells after SCI. **a** Illustration of transgenic mice used. *Myrf* ICKO and control mice were crossed with a mouse line that has a Rosa26mGFP (mT/mG) membrane-tethered GFP reporter that is Cre inducible. **b** Overview of injury epicenter at six WPI showing MBP, GFP, and NF200/SMI312 labeling. Representative areas from boxes are shown at higher magnification in **c**, **d**. **c** Single optical confocal sections stained with GFP for recombined cells, MBP to label myelin and NF-200/SMI312 to label axons in control mice.
**e** Photomicrograph of individual oligodendrocyte processes wrapping around NF-200/SMI312 + axons and colabeling with MBP in control mice (yellow arrows). **d** In *Myrf* ICKO mice, there are few MBP+ GFP+ sheaths in the ventrolateral white matter. **f** *Myrf* ICKO mice have processes that wrap NF-200/SMI312+ but these processes typically do not express MBP (blue arrow). **g** Quantification of the density of newly generated myelin sheaths (mGFP+ MBP + around NF200/SMI312+ axons) in spared tissue at two and six WPI. Control mice and *Myrf* ICKO animals do not differ at two WPI in their newly generated myelin sheath densities, but at six WPI control mice have a higher density of newly generated myelin sheaths compared to *Myrf* ICKO mice ($F_{(1,18)} = 37.77$ two-way repeated measures ANOVA, $P < 0.001$; two WPI *Myrf* ICKO vs. Control: $P = 0.812$ six WPI *Myrf* ICKO vs. Control: $P < 0.001$ Tukey's post hoc test). **h** Quantification of the percentage of MBP+ sheaths around axons that are GFP+ (new myelin) at six WPI at the lesion epicenter. There are more new myelin sheaths in control mice relative to *Myrf* ICKO ($df = 10$, $t = 10.69$, $P < 0.001$, Student's $t$ test). **i** Quantification of GFP+ processes, which completely wrap axons but fail to express detectable MBP and likely represent ensheathment by oligodendrocyte lineage cells reveals no statistical differences at six WPI ($df = 10$, $t = 1.665$, $P = 0.127$, Student's $t$ test). **j**, **k** The ventrolateral white matter at two WPI showing few GFP+ MBP+ myelin sheaths in both control animals and *Myrf* ICKO mice. ***$P \leq 0.001$, ns non-significant. Scale bar = 100 μm (**b**) and 10 μm (**c**, **d**, **j**, **k**). Error bars are mean ± SEM

of myelination. Following staining with Toluidine blue, the core of the lesion was filled with phagocytes and nearly devoid of myelinated axons at six WPI in both groups (Fig. 5a). However, in the spared ventrolateral white matter, an increasing gradient of myelinated axons radiated outwards from the lesion epicenter to the most lateral portions of the white matter (Fig. 5b). The area of spared tissue within the ventralateral white matter at the lesion epicenter was the same in each group (control 0.398 ± 0.056 vs. *Myrf* ICKO 0.396 ± 0.039 mm$^2$, $df = 8$, $t = 0.028$, $P = 0.978$ Student's $t$ test). Control mice had 27,239 ± 4587 myelinated axons relative to 15,200 ± 1616 myelinated axons in *Myrf* ICKO mice consistent with an impaired capacity to form new oligodendrocyte myelin (Fig. 5c). This reveals that *Myrf* ICKO mice had a nearly 44% decline in the number of myelinated axons suggests that upwards of 12,000 axons are typically remyelinated by oligodendrocytes within the ventrolateral white matter of mice with functional *Myrf* after moderate contusive SCI. Electron micrographs of the lesion epicenter of *Myrf* ICKO and control mice demonstrated few thinly myelinated axons, and the presence of many unmyelinated axons in *Myrf* ICKO (Fig. 5d). Quantification of the thickness of myelin relative to axon diameter revealed a shift in the distribution toward more thinly myelinated fibers in controls relative to *Myrf* ICKO's (Fig. 5e, f). G-ratios greater than 0.85 are rarely found in the uninjured rodent spinal cord[13]. The high frequency of axons with g-ratio > 0.85 in controls is likely indicative of the presence of oligodendrocyte remyelination (Fig. 5e). Lastly, there was an increase in the number of axons >1 μm in diameter that lacked myelin (1298 ± 327 axons in controls relative to 8336 ± 1072 axons in *Myrf* ICKO) indicative of profound chronic demyelination (Fig. 5g). Together, these data demonstrate that *Myrf* ICKO was effective at reducing remyelination chronically after SCI, whereas in the presence of *Myrf* there was extensive remyelination of spared axons.

**Motor recovery when oligodendrocyte remyelination is blocked**. Given the large amount of oligodendrogenesis and remyelination that occurred in control mice after moderate thoracic SCI, we wanted to understand if oligodendrocyte remyelination was causative in locomotor recovery. In contrast to mice with functional MYRF, *Myrf* ICKO mice were almost completely unable to produce new oligodendrocyte myelin resulting in profound chronic demyelination of spared axons at the lesion epicenter. Thus, *Myrf* ICKO mice provide the necessary contrast to understand the contribution of oligodendrocyte remyelination to locomotor recovery. To our surprise, we found that impaired remyelination in *Myrf* ICKO mice was not associated with a difference in functional recovery using open field

testing at any time point following SCI (Fig. 6a, b). Both *Myrf* ICKO and controls plateaued with locomotor scores of between five and a six on the BMS by six WPI, indicative of the recovery of plantar hindlimb stepping but with impairments in coordination and trunk stability. We also measured fine differences in locomotion using a regular horizontal ladder task, which is known to have higher discriminative capacity for mice with a BMS score from 5 to 7 than the BMS alone[48]. Both controls and *Myrf* ICKO mice showed an increased number of errors after injury on the horizontal ladder, however again, there was no difference between either group at any time point examined (Fig. 6c).

Mice also underwent footprint analysis using the Catwalk apparatus, which quantifies numerous aspects of gait and is capable of detecting subtle differences in locomotion[49] (Fig. 6d–f).

The base of support, stride length, relative paw position, duty cycle and the percentage of time individual paws were on the platform were analyzed. These are outcome measures sensitive to motor dysfunction following SCI[50]. In control mice and *Myrf* ICKO, injury induced profound impairments in stride length (Fig. 6g), base of support (Fig. 6h), and an increase in relative bilateral paw position (Fig. 6i), but did not alter hindlimb duty cycle in mice (time standing/time standing + time in swing) (Fig. 6j). After injury, there was also a decrease in the duration of time a mouse had one or two paws placed on the walkway (Fig. 6k) and an increase in the time in which three or four paws were simultaneously in contact with the walkway (Fig. 6l). However, we did not find at any time point a difference between between *Myrf* ICKO and controls on any of these or other parameters. These same analyses were run on an additional cohort of *Myrf* ICKO and control mice with the same protocol (Supplementary Fig. 1) and again, we found no difference in hindlimb locomotion between *Myrf* ICKO and controls using open field testing (Supplementary Fig. 1a, b), the horizontal ladder (Supplementary Fig. 1c) and on the Catwalk (Supplementary Fig. 1d–i). Combining, these separate cohorts did not result in differences between controls and *Myrf* ICKO following SCI on the BMS (Supplementary Fig. 2a), BMS subscore (Supplementary Fig. 2b), or horizontal ladder (Supplementary Fig. 2c). Importantly, when the rate of oligodendrocyte remyelination is compared to locomotor improvements, we find the majority of hindlimb recovery following moderate thoracic SCI occurs during the first two weeks, when little oligodendrocyte remyelination is present in mice (Fig. 3g relative to Fig. 6a, and summarized in Fig. 7a, b). Collectively, these data demonstrate that the initial recovery of hindlimb locomotion transpires independently of oligodendrocyte remyelination following thoracic contusive SCI.

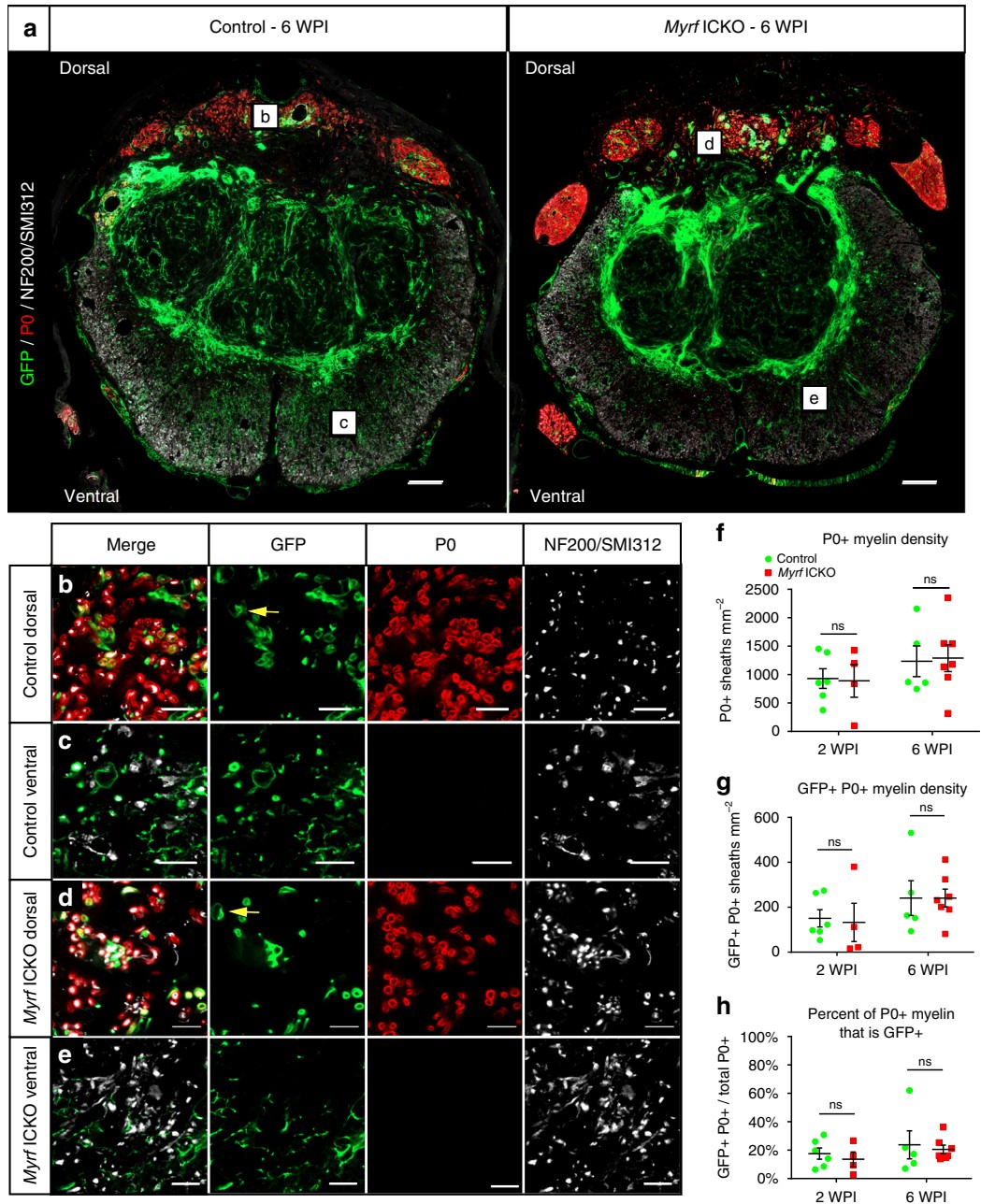

**Fig. 4** *Myrf* ICKO does not alter Schwann cell myelination following SCI. **a** Overview images of spinal cord cross sections from control mT/mG and *Myrf* ICKO mT/mG mice stained with GFP, the Schwann cell myelin marker P0 and NF-200/SMI312 to label axons. In both *Myrf* ICKO and controls, P0+ staining is mostly confined to the dorsal column. **b**–**e** Single optical confocal micrographs in either the dorsal or ventral white matter of control and *Myrf* ICKO mice with the mT/mG reporter. In the dorsal column of control and *Myrf* ICKO mice there are P0+ sheaths around NF200/SMI312+ axons, some of which colabel with GFP. There are typically very few P0+ sheaths in either the ventral white matter of control or *Myrf* ICKO mice. **f** Quantification of the total density of P0 myelin sheaths (P0+) demonstrates there is no difference between groups at two WPI ($df = 8$, $t = 0.128$, $P = 0.901$, Student's *t* test) or at six WPI ($df = 10$, $t = 0.154$, $P = 0.880$, Student's *t* test). **g** Quantification of the density of newly generated P0 myelin sheaths (P0+ mGFP+) demonstrates there is no difference between groups at two WPI ($df = 8$, $t = 0.218$, $P = 0.883$, Student's *t* test) or at six WPI ($df = 10$, $t = 0.001$, $P = 0.999$. Student's *t* test). **h** The percentage of P0+ myelin sheaths, which are derived from PDGFRα+ cells relative to the total P0+ myelin sheaths do not differ between knockouts and controls at two WPI ($df = 8$, $t = 0.621$, $P = 0.552$. Student's *t* test) or at six WPI ($df = 10$, $t = 0.364$, $P = 0.724$, Student's *t* test). ns =non-significant. Scale bar = 100 μm (**a**), 10 μm (**b**–**e**). Error bars are mean ± SEM

## Discussion

Myelin regeneration is considered a key therapeutic target to enhance function following SCI, but the transcriptional control and functional relevance of this process are unknown. We used a loss-of-function approach to ascertain the role of endogenous oligodendrocyte remyelination in locomotor improvements following SCI. By removing *Myrf* from PDGFRα+ OPCs, the accumulation of new oligodendrocytes was largely inhibited, while OPC proliferation and recruitment in response to SCI were preserved. *Myrf* ICKO blocked oligodendrocyte remyelination

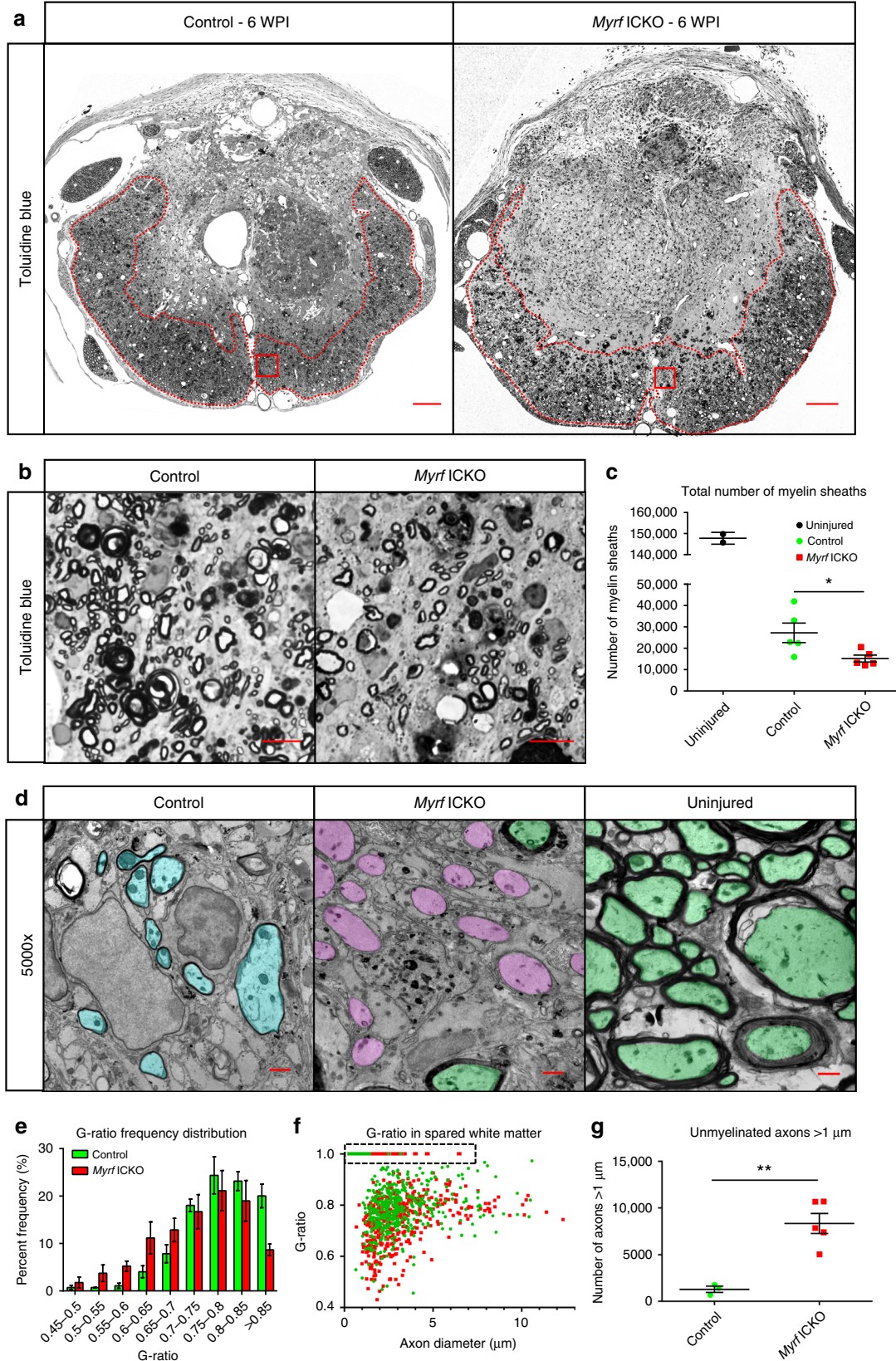

from recombined cells, resulting in a 44% decline in the number of myelinated axons at the lesion epicenter in *Myrf* ICKO mice. Surprisingly, despite chronic demyelination, knockout of *Myrf* from OPCs did not alter the amount or rate of hindlimb motor recovery in this model. Further, by genetically fate mapping new

myelin formation we demonstrated that the vast majority of oligodendrocyte remyelination occurs after the recovery of hindlimb stepping. Therefore, oligodendrocyte remyelination is not a crucial component to recovery of hindlimb stepping following moderate thoracic SCI in mice.

**Fig. 5** Chronic demyelination of spared axons in *Myrf* ICKO six weeks following SCI. **a** Whole cross sections of control and *Myrf* ICKO spinal cords at lesion epicenter stained with Toluidine blue at six WPI. The majority of myelin is found in the ventrolateral white matter. **b** High magnification images of box inset from **a** in *Myrf* ICKO and control animals. **c** Quantification of myelinated axons in the spared white matter. *Myrf* ICKO animals have significantly fewer myelinated axons when compared to control animals ($df = 8$, $t = 2.475$, $P = 0.038$, Student's $t$ test). **d** Example transmission electron micrographs of the injured mouse lesion epicenters. Blue shading depicts thinly myelinated axons, pink shading depicts axons devoid of myelin, and green shading depicts axons with thick myelin sheaths. Many thinly myelinated axons are found in control mice whereas *Myrf* ICKO mice are almost completely devoid of thinly myelinated large caliber axons, and instead have demyelinated axons greater than 1 μm in size at six WPI. **e** Frequency distribution of g-ratios of myelinated axons indicate a shift towards higher g-ratios (more thinly myelinated axons) in the controls relative to *Myrf* ICKO ($P < 0.001$, Kolmogorov–Smirnov test). **f** Scatter plot comparing g-ratio to axon diameter of axons quantified in the spared white matter of injured animals demonstrating a difference between controls and *Myrf* ICKO ($F = 25$, $DFn = 1$, $DFd = 1340$, $P < 0.0001$, linear regression). Dashed box highlights axons that lack myelin. **g** Quantification showing more unmyelinated axons larger than 1 μm in the spared white matter of *Myrf* ICKO compared to controls at six WPI ($df = 6$, $t = 4.858$, $P = 0.003$, Student's t-test). $^*P \leq 0.05$, $^{**}P \leq 0.01$ Scale bars = 100 μm (**a**), 5 μm (**b**), 1 μm (**f**). Error bars are mean ± SEM

We demonstrate the expression of MYRF in PDGFRα+ cell-derived oligodendrocytes is essential for effective remyelination of the spinal cord following traumatic injury. *Myrf* deletion from OPCs did not affect OPC recruitment or proliferation. Microarray and immunohistochemical stains demonstrate that MYRF is not typically expressed in OPCs[31,32], so it is not surprising their density or proliferation is not altered after SCI. Emerging evidence suggests OPCs may be crucial for mediating inflammation[51] and may entrap dystrophic axon tips within the glial scar[52], so altering OPC numbers would confound an interpretation of the role of oligodendrocyte remyelination in locomotor recovery. Reduced oligodendrogenesis in *Myrf* ICKO mice were likely a result of a failure of OPCs to fully differentiate/mature and subsequently being more vulnerable to apoptosis[31]. Oligodendrogenesis by resident PDGFRα+ OPCs cannot be compensated for by other cell sources like ependymal cells or Schwann cells, even when resident OPC differentiation is blocked following SCI. Therefore, PDGFRα+ progenitors have an essential role in the generation of new oligodendrocyte myelin after SCI, analogous to their role in chemical demyelination[53]. Blocking the accumulation of new oligodendrocytes reveals a large number of axons at the lesion epicenter (~12,000) are normally receptive to oligodendrocyte remyelination after moderate thoracic SCI consistent with extensive early demyelination[8,9]. However, when mice have functional *Myrf*, most of these axons are remyelinated by six weeks post-SCI. Therefore, we demonstrate that PDGFRα+ OPCs require MYRF expression upon differentiation and have an indispensable role in generating new oligodendrocytes to remyelinate after SCI.

Given the high level of endogenous remyelination that occurs following SCI, it is surprising that hindlimb motor recovery is not affected when oligodendrocyte remyelination is ablated. One possibility is that myelin formed in response to SCI fails to restore conduction. While we cannot directly discount this possibility, the increased conduction velocity seen with the onset of remyelination at 2 weeks[13], and computer modeling indicating very thin myelin is sufficient to improve conduction[54] argues against such conduction failure. In cats fed an irradiated diet during gestation myelin vacuolation and severe demyelination are observed[34]. When these cats are returned to a normal diet they generated thin myelin that is restorative for function[34], indicating that thinly remyelinated axons in the spinal cord are functional, at least in this case. A second, and in our view more plausible alternative, is the limited rostral–caudal extent of demyelination along axons may not be sufficient to block conduction long-term. Segmentally demyelinated spinal cord axons can conduct in vivo through demyelinated lengths of at least 2.5 mm[41], and following SCI, demyelination of spared axons has been shown to be reasonably focal to the lesion epicenter[40]. Labeling of descending rubrospinal tract axons after contusion injury indicates that 80% of the abnormally short internodes (<100 μm), suggestive of new adult-generated myelin, are found within 1 mm rostral and caudal of the lesion[40]. Conduction can be restored in demyelinated axons by the redistribution of sodium channels along the demyelinated axolemma[55], a process that may take time, but could explain the partial restoration of conductance at one and two WPI prior to extensive oligodendrocyte remyelination[13]. Thus, perhaps over short distances of demyelination like those observed in rodent SCI, remyelination is not required to activate residual neural circuitry.

Locomotor recovery following incomplete SCI relies on the reorganization of descending circuits to deprived spinal segments[56,57], and to changes in cellular and circuit properties within the central pattern generator[58,59], and motor neurons[60] below the level of injury. In cervical models of SCI, very few corticospinal neurons can mediate forelimb motor improvements[5,61], and sparing of less than 20% of the ventrolateral funiculus is associated with locomotor recovery following thoracic SCI[62,63]. As such, relatively few descending circuits may be necessary to reestablish the excitatory input required for locomotor recovery[5,62,63]. Given this, remyelination of descending axons may not be functionally relevant except in cases where very few axons persist[62], or more sustained/extensive demyelination is observed.

Schwann cell myelination was observed within the first 2 weeks following SCI when the majority of locomotor recovery occurs. Schwann cell myelin within the CNS is sufficient to improve conductance following CNS demyelination[64], and transplantation of Schwann cells into the injured spinal cord has been reported to confer functional benefits[42]. Consistent with the possibility of Schwann cells potentially driving a portion of recovery is a recent study demonstrating that the inducible knockout of neuregulin-1, which prevents Schwann cell myelination following moderate thoracic SCI, is correlated with diminished functional locomotor recovery[27]. Importantly, we found Schwann cell myelination, in contrast to oligodendrocyte remyelination, occurs early enough after injury to potentially mediate recovery. However, determining the role of Schwann cell myelination during recovery following SCI still requires future cell-specific knockout experiments.

Interestingly, the ablation of oligodendrocyte remyelination did not induce a compensatory increase in Schwann cell myelination, which was still primarily confined to the dorsal column. This raises the intriguing possibility, that the injury environment leaves different CNS axon populations selectively permissible to either Schwann cell or oligodendrocyte remyelination following SCI. The size of the axon[24], and the proximity to peripheral roots[65] may be factors contributing to Schwann cell generation from OPCs, but astrocytes seem to have the prominent role in regulating the level of Schwann cell myelination[66,67]. Schwann cell myelination is confined to areas depleted of astrocytes after SCI[66], and STAT3-mediated reactive astrogliosis restricts

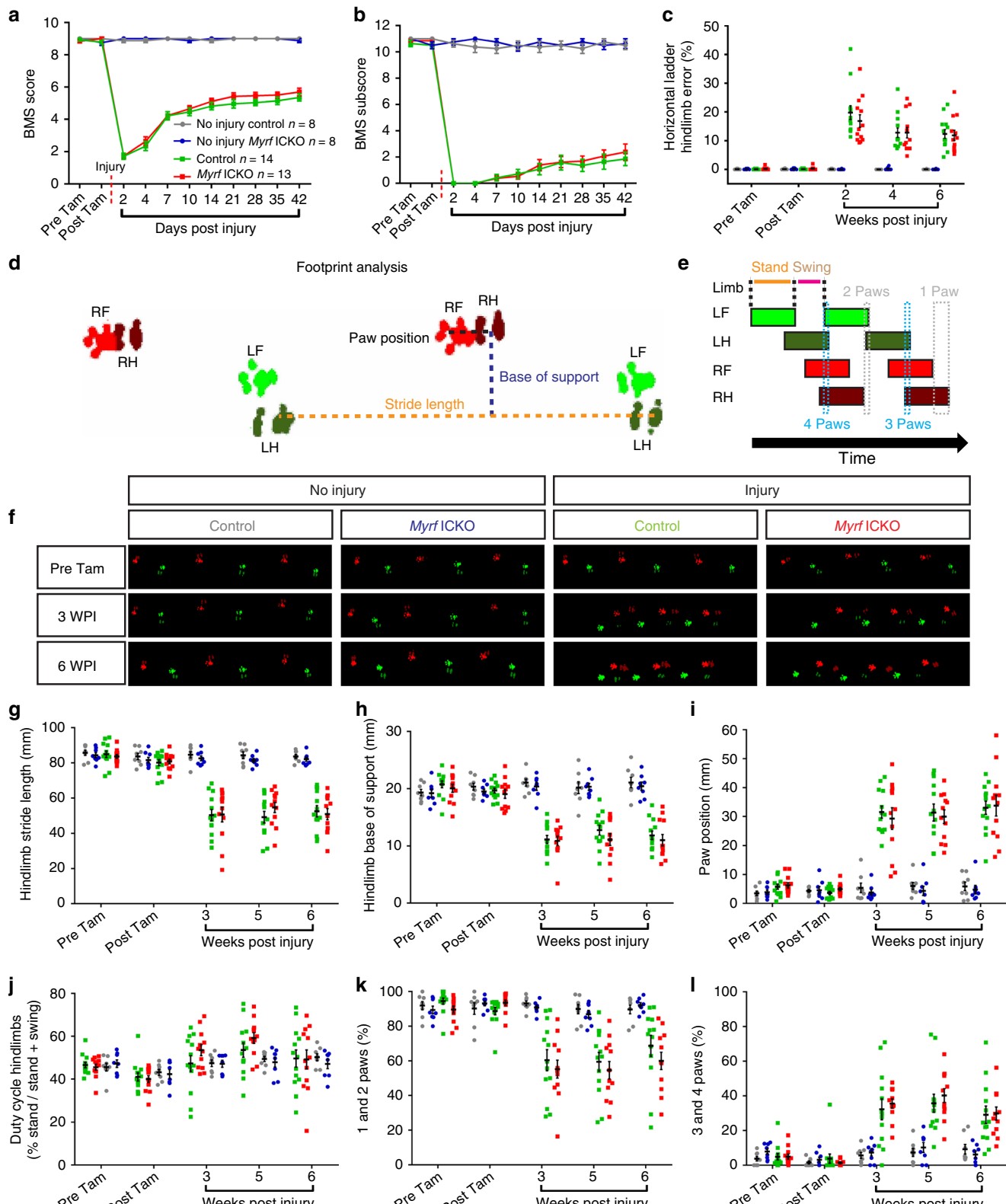

Schwann cell myelination following demyelination[67]. The knockout of *Myrf* from OPCs does not alter astrocyte coverage in the injured spinal cord, which may have restricted a compensatory increase in Schwann cell myelination.

Remyelination has been considered a promising target for SCI[16–19]. Oligodendrocyte remyelination is extensive by six WPI, but nearly absent at two WPI when hindlimb stepping has typically recovered following moderate thoracic SCI. It is conceivable that an acceleration of oligodendrocyte remyelination within the first 2 weeks after SCI could speed or increase the extent of locomotor recovery. In accordance, cell transplants targeting remyelination injected within the first 2 weeks were

**Fig. 6** *Myrf* deletion from PDGFRα+ cells does not impair motor recovery following moderate thoracic contusive SCI. **a** Time course of locomotor function evaluated by open field BMS. While *Myrf* ICKO and controls did not differ after SCI ($F(3, 39) = 286.0$, $P < 0.001$; injured *Myrf* ICKO vs. control $P = 0.518$), both SCI groups were statistically different from uninjured controls at all time-points postinjury ($P < 0.001$). **b** On the BMS subscore, there is no difference between *Myrf* ICKO and controls ($F(3, 39) = 388.0$, $P < 0.001$; injured *Myrf* ICKO vs. control $P = 0.966$). **c** There is no difference between *Myrf* ICKO and controls in the percentage of errors (error/error + success) on the horizontal ladder after SCI ($F(3, 38) = 25.86$, $P < 0.001$, injured *Myrf* ICKO vs injured control Tukey's post hoc P=0.942). **d** An illustration of paw recordings from the Catwalk along with parameters in **g**–**i** used to assess gait. LH left hindlimb; LF left forelimb; RF right forelimb; RH right hindlimb. **e** Example of the time course in which a paw is in contact with platform (colored boxes). **f** Example recordings of three full step cycles from the Catwalk prior to injury and tamoxifen dosing, at three WPI, and at six WPI. **g**–**l** No differences in gait were observed between *Myrf* ICKO and controls either with or without an injury on **g** hindlimb stride length ($F(3, 37) = 44.13$, $P < 0.001$; injured *Myrf* ICKO vs. injured control $P = 0.977$). **h** Hindlimb base of support ($F(3, 37) = 48.09$, $P < 0.001$; injured *Myrf* ICKO vs. injured control $P = 0.630$). **i** Combined paw position ($F(3, 37) = 52.74$, $P < 0.001$; injured *Myrf* ICKO vs. injured control $P = 0.983$). **j** Hindlimb duty cycle ($F(3, 37) = 0.933$, $P = 0.435$; injured *Myrf* ICKO vs. injured control $P = 0.738$). **k** Percent of run with one or two paws on the platform ($F(3, 37) = 17.47$, $P < 0.001$; injured *Myrf* ICKO vs injured control $P = 0.651$). **l** Three or four paws on the platform ($F(3, 37)$ 15.46. $P < 0.001$; injured *Myrf* ICKO vs. injured control $P = 0.934$). Groups were compared at all post injury time points. All statistical comparisons were made using a two-way repeated measures ANOVA, and a Tukey's post hoc for individual group differences. Error bars are mean ± SEM

associated with more remyelination and improved locomotor recovery whereas more chronic transplants did not alter remyelination or subsequent locomotor recovery[35,37]. However, both the time course of oligodendrocyte remyelination, and the unimpaired recovery relative to control mice in the absence of oligodendrocyte remyelination question both the role of oligodendrocyte remyelination in this model to drive recovery and the viability of this model to test remyelinating therapies. Ultimately, this study raises doubts whether remyelination is a validated target for clinical translation following moderate spinal cord contusion.

## Methods

**Transgenic mice.** Procedures involving live animals were approved by the University of British Columbia, in accordance with guidelines from the Canadian Council on Animal Care (A13-0328). Experiments were initiated in 8–10-week old mice that were group housed, fed a standard chow *ad libitum* and maintained on a 12 h reverse dark/light cycle for the experiment. Mice from the parental *Myrf*[fl/fl] line (Jackson Laboratory stock # 010607)[32], which express LoxP sites around both copies of exon 8 of *Myrf* (homozygous), were crossed with PDGFRα-CreERT2[44] (Jackson Laboratory stock # 018280) mice. Offspring from the F1 generation with the PDGFRα-CreERT2 transgene and heterozygous for the presence of LoxP sites around *Myrf* (*Myrf*[fl/wt]) were crossed with F1 *Myrf*[fl/wt] mice lacking the PDGFRα-CreERT2 transgene. F2 generation *Myrf*[fl/fl] PDGFRα-CreERT2 and *Myrf*[fl/fl] mice lacking the Cre transgene were then bred to produce sufficient mice for the experiments. This breeding strategy yielded litters in which all mice had both copies of *Myrf* surrounded by LoxP sites (*Myrf*[fl/fl]) with individual mice either without (control mice) or with the PDGFRα-CreERT2 transgene (*Myrf* ICKO). All *Myrf* ICKO mice used were heterozygous for PDGFRα-CreERT2. Mice were on a mixed strain background comprised of C57bl/6 and SJL. Exon 8 of *Myrf* contains the putative DNA binding domain and its deletion results in a truncated, non-functional protein[31,32,45]. All animals, unless stated otherwise, were treated with tamoxifen to ensure no confounding effects of the drug on either recovery from SCI, or remyelination efficiency. The insertion of the PDGFRα-CreERT2 transgene does not affect recovery following contusive SCI relative to *Myrf*[fl/fl] lacking the PDGFRα-CreERT2 in the absence of tamoxifen (Supplementary Fig. 3).

To determine the extent of new myelin produced by OPCs, *Myrf*[fl/fl] PDGFRα-CreERT2 or *Myrf*[wt/wt] PDGFRα-CreERT2 mice were crossed with Rosa26-mGFP (mT/mG) mice (JAX # 007576)[68], which induces GFP expression that is tethered to the membrane following tamoxifen induced Cre-mediated recombination (mGFP). *Myrf* ICKO and control mice were heterozygous for the mT/mG and PDGFRα-CreERT2 transgenes in genetic fate mapping experiments. A total of $n = 23$ mice were used in the study and $n = 1$ animal died during surgery bringing the total animals to $n = 12$ perfused at six WPI and $n = 10$ perfused 2-week postinjury. Genotyping was performed on ear clippings and DNA was extracted using the REDExtract-N-AMP Tissue Kit (Sigma, St. Louis, MO, R4775) and amplified with primers specific for the transgenes[32,44,68]. Genotypes were visualized before and after the experiment by running PCR solutions on a 1.5% agarose (Invitrogen, Carlsbad, CA, 16500) gel. Primer sequences were:

  *Myrf* forward: AGGAGTGGTGTGGGAAGTGG
  *Myrf* reverse: CCCAGGCTGAAGATGGAATA
  PDGFRα CreERT2 forward: TCAGCCTTAAGCTGGGACAT
  PDGFRα CreERT2 reverse: ATGTTTAGCTGGCCCAAATG
  Rosa26-mGFP (mT/mG) common forward: CTCTGCTGCCTCCTGGCTTCT
  Rosa26-mGFP (mT/mG) wildtype reverse: CGAGGCGGATCACAAGCAATA
  Rosa26-mGFP (mT/mG) mutant reverse: TCAATGGGCGGGGGTCGTT

**Experimental design.** A total of $n = 76$ mice were used in two cohorts. The larger cohort was used for histological analysis, and behavioral scores were reported for this cohort in the results. Cohorts were not combined as injuries were done on two different IH impactors, and resulted in slightly different levels of recovery between the control groups beginning three WPI until six WPI ($F(1, 21) = 9.850$, two-way repeated measures ANOVA, $P = 0.005$). Behavioral data from the second cohort was reported in Supplementary Fig. 1, and combined data in Supplementary Fig. 2. Group sizes were determined prior to experiment by conducting a power analysis from data generated in a pilot experiment with mice on the same genetic background to determine the group size which is required to detect a 1 BMS difference ($n = 14$ per group) given the variability in our data ($\alpha < 0.05$, Power = 0.80). To ensure we had sufficient power, a total of $n = 60$ *Myrf* ICKO and control mice received spinal cord injuries. Animal grouping was dependent on genotype. An additional, $n = 16$ (8 *Myrf* ICKO and 8 controls, split evenly between males and females) aged-matched mice without an injury were examined to determine if *Myrf* ICKO was sufficient to induce demyelination or behavioral deficits during the timeframe without an injury. $n = 9$ injured animals ($n = 4$ *Myrf* ICKO, 5 controls) were excluded due to subsequent health issues including digit autotomy (1 mouse), hernia (1 mouse), bladder infections/complications (5 mice), or surgical deaths (2 mice). Additionally, mice were excluded after the experiment if they were statistical outliers (below lower quartile $-1.5\times$ the interquartile range or above the upper quartile + $1.5 \times$ interquartile range) on displacement relative to their experimental grouping (2 mice) or if they demonstrated evidence of plantar stepping immediately post injury (3 mice) (BMS score $\geq 4$), both a priori exclusion criteria. The remainder of the mice ($n = 46$) were used in behavioral and histological analyses and there were $n = 23$ *Myrf* ICKO and $n = 23$ controls, with $n = 12$ males in the control group and $n = 10$ males in the *Myrf* ICKO group. Mice were perfused for use in either immunohistochemical or electron microscopic analysis. Mice used for electron microscopy were grouped so they did not statistically differ in their BMS scores from those used for immunohistochemistry.

**Spinal cord injury and animal care.** Prior to surgery, mice were anaesthetized for 3 min with a 3% isofluorane (Fresenius Kabi, Toronto, Canada, CPO40602) to oxygen mixture. Anesthesia was maintained at 1.5–2% isofluorane as needed during surgery. Each animal received 1 ml of Ringer's solution (Braun, Montreal, Canada, L7500) and buprenorphine (0.05 mg/kg) (Reckitt-Benckiser Slough, Toronto, Canada) analgesic prior to surgery. The back was shaved and then disinfected using successive betadine (Purdue Pharma, 41731) and 70% alcohol washes. An incision and separation of the erector trunci muscles from the spine followed by a dorsal laminectomy of T9–10 was performed. The vertebral column was stabilized by clamping the exposed T8 and T10 vertebrae with forceps prior to positioning the animal under the Infinite Horizons Impactor[69] (Precision Systems). The IH impactor tip was lowered until it just contacted the exposed spinal cord, raised 1 cm, and set to deliver 70 kilodynes of force. Following surgery, the skin and overlying musculature were sutured with 6-0 nylon sutures (Ethicon, San Lorenzo, Peurto Rico, 667G) and the mice were placed into a temperature and humidity controlled incubator at 32 °C until they awoke. Mice were administered buprenorphine twice daily for the following 2 days and Ringer's solution daily for 5 days or longer if needed. Bladders were also expressed twice daily until spontaneous micturition was achieved.

**Tamoxifen and EdU administration.** Tamoxifen was dissolved in corn oil (Sigma, St. Louis, MO, C8267) at 20 mg ml$^{-1}$ before administration. All mice received 100 mg kg$^{-1}$ day$^{-1}$ intraperitoneal injections of tamoxifen (Sigma, St. Louis, MO, T5648) beginning 9 days prior to SCI and continuing for 5 consecutive days. For the first two days after SCI, 5-ethynyl-2'-deoxyuridine (EdU) (Invitrogen, Eugene, OR A10044) was dissolved in sterile PBS and administered by intraperitoneal injection (5 mg kg$^{-1}$). After two days, EdU (Carbosynth, San Diego,

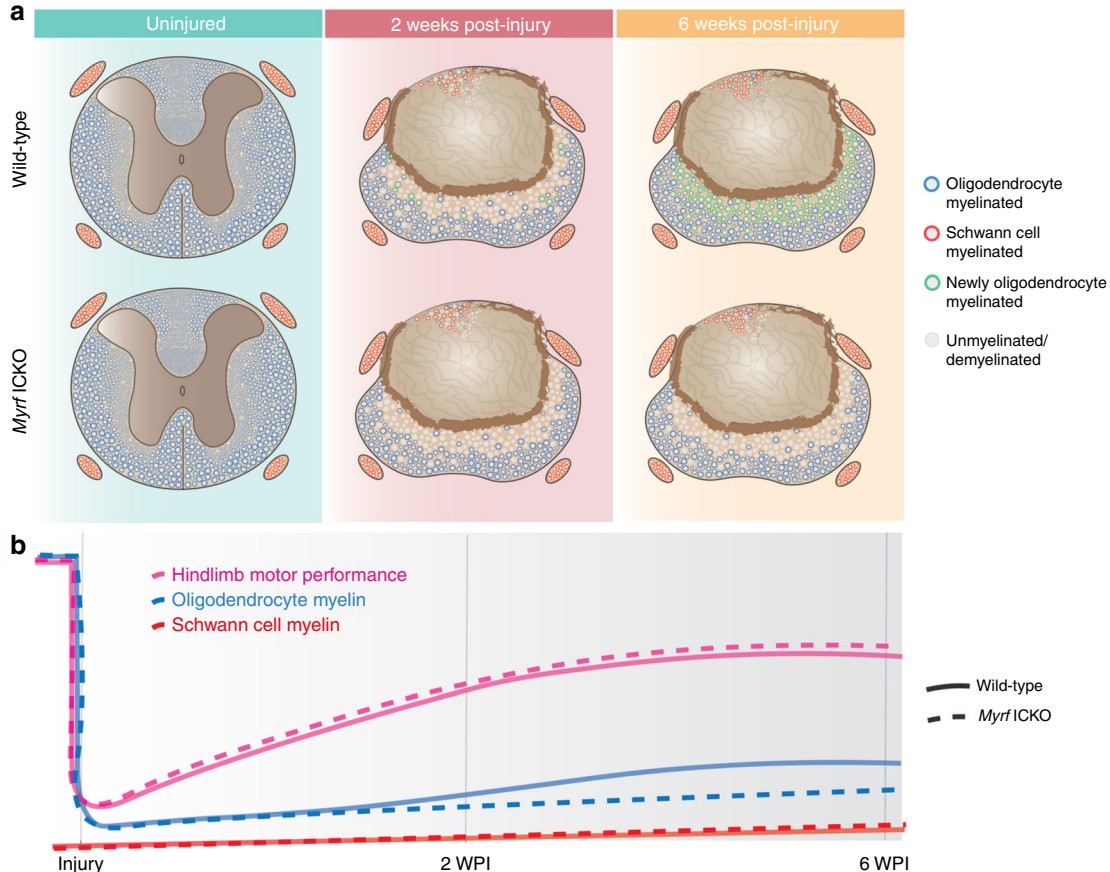

**Fig. 7** New oligodendrocyte and Schwann cell myelination after SCI and its relationship to locomotor recovery. **a** Schematic of the uninjured and injured mouse spinal cord two and six WPI following moderate dorsal thoracic contusion. In the uninjured spinal cord, axons are myelinated solely by oligodendrocytes and peripheral nerves are myelinated by Schwann cells. By two WPI, the lesion epicenter is ringed by a glial scar and mostly devoid of axons. Demyelinated axons are seen in a gradient increasing outwards from the medial spinal cord. By six WPI, extensive oligodendrocyte remyelination is observed throughout the ventrolateral white matter in control mice with functional MYRF, but *Myrf* ICKO mice fail to produce new oligodendrocyte myelin. Schwann cell myelination is generally confined to the dorsal column. The degree of Schwann cell myelination does not differ in the injured spinal cord between *Myrf* ICKO and control mice. **b** Diagram illustrating the relative amount and rate of open field hindlimb motor performance compared to the extent of oligodendrocyte and Schwann cell myelin after injury in the spinal cord. After thoracic SCI, there is a decline in both hindlimb motor performance and number of myelinated axons in the CNS. The majority of recovery of hindlimb locomotor function on open field testing occurs within the first two weeks in both *Myrf* ICKO and controls. In contrast, the vast majority oligodendrocyte remyelination does not occur until after two weeks postinjury. Therefore, the relative time course of oligodendrocyte remyelination is not associated with hindlimb motor recovery after SCI. In contrast, Schwann cell myelination occurs within the first two weeks after SCI and occurs at a relatively steady rate. The height of the lines is approximately proportional to the extent of loss and subsequent recovery after SCI

CA, 61135-33-9) was dissolved in drinking water (0.2 mg ml$^{-1}$) with 1% D-glucose to encourage consumption[46]. EdU water was changed every 2 days and the mice were administered EdU in their water until four WPI.

**Perfusion and tissue processing**. To collect spinal cords for immunohistochemical analysis, mice were transcardially perfused with 20 ml of PBS followed by 40 ml of freshly prepared 4% paraformaldehyde (PFA) (Fisher Scientific, Ward Hill, MA A11313) at two or six WPI. The injury site was identified, then one cm of the spinal cord flanking the injury was dissected. Spinal cords were fixed in PFA for 8 h, then incubated in ascending sucrose solutions (12, 18, and 24%) at 4 °C. Tissue was submerged in OCT compound (Tissue-Tek, Torrance, CA 4583) frozen on dry ice and stored at −80 °C. All spinal cords were sectioned using a cryostat (Thermo Scientific, Walldorf, Germany, HM-525) into 20 μm thick cross-sections, which were mounted in series on ten slides making each individual section on a slide 200 μm apart.

Spinal cords were collected for electron microscopy at six WPI. Mice were transcardially perfused with 20 mL of 0.01 M PBS followed by 40 ml of 4% PFA with 1% glutaraldehyde chilled to 4 °C (Electron Microscopy Sciences, Hatfield, PA, 16220). The injury site was identified, then segments of the spinal cord were removed at, and adjacent to, the lesion epicenter. The epicenter and adjacent sections were dissected into 1 mm blocks and fixed in 2% glutaraldehyde for 2 h before being washed three times in 0.1 M cacodylate buffer with 5.3 mM CaCl₂, and

then incubated with 1% osmium tetroxide (Electron Microscopy Sciences, Hatfield, PA, 19190) with 1.5% potassium ferrocyanide (BDH, Toronto, Canada) for 1.5 h. Once fixed, the tissue went through ascending alcohol washes before being washed with propylene oxide (Electron Microscopy Sciences, Hatfield, PA 20401) and embedded in Spurr's resin (Electron Microscopy Sciences, Hatfield, PA 14300).

**Immunohistochemistry**. To prepare for antibody staining, slides were thawed then rehydrated in PBS. In order to effectively stain myelin proteins, tissue was put through ascending, then descending ethanol dilutions (50, 70, 90, 95, 100, 95, 90, 70, 50%), followed by three washes of PBS. Tissue was then blocked with 10% normal donkey serum dissolved in PBS with 0.1% Triton X-100 for 30 min. Primary antibodies were diluted in PBS with 0.1% Triton X-100 and applied to the slides overnight at room temperature in a humid chamber. The following morning, slides were washed and incubated with donkey Dylight or Alexa Fluor secondary antibodies (Jackson ImmunoResearch Laboratories, Inc.) for 2 h, then washed again before being coverslipped using Fluoromount-G (Southern Biotech, 0100-01). Antibodies used were raised against the following antigens: CC1 (1:300, Millipore, OP80), OLIG2 (1:500, Millipore, AB9610), MYRF (1:300, N-terminus, generously provided by Dr. Michael Wegner), GFP (1:4000, Abcam, ab13970), GFAP (1:1000, Sigma, G3893), PDGFRα (1:200, R and D Systems, AF-307-NA), NF200 (1:1000, Sigma, N0142), SMI312 (1:1000, Covance, SMI-312R-100) and P0 (1:100, Aveslabs, PZO).

**Cell counting and tissue analysis**. All analyses were performed blinded to animal genotypes. A Zeiss Axio-Observer M1 inverted confocal microscope with a Yokogawa spinning disk and Zen 2 software (Zeiss) was used for imaging. For analysis of the area of spared tissue, images of whole spinal cord cross sections stained with GFAP were taken at 100× magnification and analyzed in ImageJ (NIH). The intact area was determined by manually circling the lesion border indicated by GFAP + immunoreactivity with spared cytoarchitecture then calculating the total area for each section.

For analysis of cell densities, we imaged the epicenter of injury and the next two sections 200 and 400 μm rostral and caudal for each animal for a total of five sections per mouse. We performed systematic uniform random sampling within each section[70] by overlaying a grid (individual grid size 103 μm × 108 μm) onto a low magnification preview image of a cross section of spinal cord. One counting square for every 3 × 3 grid area was imaged at 400× magnification. Z-stacks were imaged through the entire depth of the 20 μm thick section with 1 μm spacing between optical sections, and cells were counted in three dimensional space within a 100 × 100 μm optical disector. Nuclei that came into focus and were within the optical disector or in contact with right and upper edge were counted to ensure only unique objects were quantified. We analyzed approximately 10–15 Z-stacks per spinal cord section, depending on the size of the cord after injury. The number of cells within the specified volume of the sampling box were averaged per section, giving the density of cells per $mm^3$.

Assessments of the density of newly generated myelin sheaths were examined within the spared tissue at epicenter and 200 μm rostral and caudal. Similar to the quantifications for cell densities, we overlaid a grid (individual grids 68.5 μm × 69.0 μm) across a cross section of the spinal cord then used systematic uniform random sampling to image a Z-stack in one out of every nine 3 × 3 grid boxes at 630× magnification through its entire depth. We imaged approximately 15–20 images per spinal cord cross section, depending on the size of the cord after injury. Images were quantified in the middle optical section of the in focus Z stacks within an optical disector of 4047.4 $μm^2$. New myelin was defined as colocalization of GFP fluorescence with myelin fluorescence (MBP or P0) that fully surrounded an axon (SMI312 or NF200-positive).

**Toluidine blue staining and electron microscopy**. Spinal cords embedded in resin were sectioned to 1 μm thickness on an ultramicrotome (Ultracut E, Reichert-Jung). Ultra- and semithin sections were collected every 20 μm and semithin sections were viewed under a light microscope to find the injury epicenter, defined by the lowest number of myelinated axons by a rater blinded to genotype. Myelin was visualized in semithin sections by brief staining with 1% toluidine blue and 2% borax solution then coverslipped with Permount (Fisher Scientific, Fair Lawn, NJ, SP15). The imaging of toluidine blue semithins was performed on a Zeiss, Axio Imager.M2 microscope at 630× magnification. The entire cross section at epicenter was imaged. A grid with box dimensions of 50 μm × 50 μm was overlaid on top of the merged image. We employed systematic uniform random sampling, counting one in every seven grids. This was done over the extent of the spared ventrolateral white matter at the injury epicenter. Spared tissue was indicated by intact cytoarchitecture and the presence of myelin sheaths. Between 1500 and 2500 myelin sheaths were counter per animal.

For transmission electron microscopy, ultrathin sections of 100 nm thickness at the lesion epicenter were stained with Reynold's lead citrate and uranyl acetate to enhance contrast, and imaged at 5000× primary magnification on a Zeiss EM910 equipped with a digital camera. At least ten nonoverlapping electron micrographs were systematically imaged within the dorsal column and spared ventrolateral white matter at six WPI. Myelin, axon diameter, and axon density were determined by a blinded observer. G-ratios were evaluated from a total of 533 axons in control, and 811 axons in *Myrf* ICKO mice. The density of unmyelinated axons was multiplied by the area of intact white matter to determine the total number of unmyelinated axons for each mouse at lesion epicenter.

**Behavioral assessments**. All behavioral assessments were performed during the dark cycle to increase activity. The raters were blinded to animal genotype while running behavioral tests and during subsequent analyses. Behavioral assessements were run in mice lacking the mT/mG reporter to avoid any confounding effects of the expression of fluorescent proteins on myelin compaction or behavioral function. Three different motor behavioral assessments were conducted: open field testing using the Basso mouse scale (BMS)[69], regular horizontal ladder[48], and Catwalk gait analysis[50]. Mice were handled repeatedly and pretrained on the Catwalk and horizontal ladder taks by running the mice three times per day for five consecutive days prior to baseline testing. Mice were familiarized with BMS open field box with cagemates, then alone prior to testing. All animals were then tested before and after tamoxifen induction to establish baseline values.

During open field BMS testing mice were placed into a 150 by 90 cm clear plexiglass box with 30 cm high sides and observed by two blinded raters. The BMS assessed hindlimb function, tail position, trunk stability, and coordination[69]. Scores were averaged between limbs. The BMS subscore, a cumulative score based off the frequency of stepping, paw position, level of coordination, trunk stability and tail position, was also recorded[69].

For horizontal regular ladder analysis, mice were videotaped with a high definition camera (Sony, HDR-XR200) crossing of 30 rungs spaced 1.3 cm apart at

30 cm height. Each mouse had five complete runs recorded and analyzed per time point. Mice were rerun if they paused for more than several seconds, reared or reversed course. At least 15 min were given per mouse between runs to reduce fatigue. Analysis reported the number of success (plantar or toe placements on the rung or skipped rungs) as well as errors (slips, misses, and drags) and scored by a blinded observer[48]. Three mice were excluded from analysis ($n = 2$ controls and $n = 1$ *Myrf* ICKO). These mice dragged more than half the run during the two WPI time point and were strong outliers for percent error (three times the interquartile range from the first or third quartile) at two or more consecutive time points after injury.

Gait analysis was performed using the Noldus Catwalk. The camera was placed 20 cm below the runway and mice run through a 5 cm wide darkened tunnel. At least five uninterrupted crossings with continuous movement were recorded. The runs were averaged for each animal. Only runs which had three consistently sped step cycles were analyzed (at least four per animal per timepoint). The settings on the Catwalk were contrast 3990, brightness −140 mV, and analyzed at gain of 10. During weeks three and five, the catwalk brightness was increased to −80 mV and analyzed at gain of 16 to better resolve footprints in animals with poor gait. As this could affect the intensity score this analysis was not compared between time points.

**Statistical analysis**. Statistical analyses were conducted using the Statistical Package for the Social Sciences (SPSS) or Graphpad 6.0 (Prism). Individual data points were displayed when possible and represent a single mouse. However, bar graphs were plotted for lesion size, the contribution of PDGFRα-cell derived myelin to total myelin, g-ratio frequency and BMS scores to increase clarity of the data. If data met assumptions for normality, tested with the Shapiro–Wilk test, t-tests were run with or without Welch's correction depending on the homogeneity of variance (tested with Levene's test). Comparisons of the density of recombined oligodendrocytes at specific distances from lesion epicenter or lesion area were compared using a two-way ANOVA with Tukey's post hoc test to detect individual differences. Linear regression was used to determine differences between g-ratios across the range of axon diameters. The Kolmgorov–Smirnov test was used to determine differences in the percent distribution of g-ratios between control and *Myrf* ICKO mice. For behavioral analyses and analyses, a two-way repeated measures ANOVA was conducted with comparisons using Tukey's or Šidák post hoc to compare individual groups. Comparisons were two-tailed and considered statistically significant if P values (P) < 0.05.

**Data availability**. All relevant data and step by step procedures used in this study are available from the authors by request.

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

## Acknowledgments

The *Myrf*<sup>fl/fl</sup> line used in this study was a generous gift of Drs. Ben Barres and Ben Emery. The PDGFRα CreERT2 line used in the study was a kind gift of Dr. Dwight E. Bergles and Dr. Shin H. Kang. The authors would like to thank Phillip Chau and Sheida Naderi-Azad for their assistance with motor behavioral analysis. Additionally, we would like to thank Susan Shin for her technical expertise with electron microscopy. We would like to thank Michael J. Lee, Akash Gupta, and Maia Bloomberg for their assistance with animal breeding, genotyping, and husbandry. We appreciate Kathleen Kohlamainen's help with behavioral analyses. This work was funded by the Canadian Institute of Health Research (CIHR) operating grant (MOP-130475). G.J.D. is supported by a Multiple Sclerosis Society of Canada Doctoral Scholarship and S.B.M. by a Multiple Sclerosis Society of Canada master's award. B.J.H. received a CIHR doctoral award and P.A. received a Frederick Banting and Charles Best Canadian Graduate Scholarship-Doctoral Award. J.R.P. was supported by the Donna Joan Oxford Postdoctoral Fellowship Award from the Multiple Sclerosis Society of Canada and postdoctoral fellowship awards from CIHR, Alberta Innovates Health Solutions and T. Chen Fong and Alberta Initiatives Health Solutions. W.T. holds the John and Penny Ryan British Columbia Leadership Chair in Spinal Cord Research.

## Author contributions

G.J.D., S.B.M., J.R.P., and W.T. envisioned and designed the experiments. G.J.D., S.B.M., and B.J.H. performed motor behavioral assessments. P.A. and J.L. assisted with surgeries and mouse experiments. G.J.D., S.B.M., and A.M. conducted histological analysis. G.J.D., S.B.M., B.J.H., J.R.P., and W.T. wrote the manuscript and all authors edited the manuscript. W.T. supervised the experiments.

## Additional information

**Competing interests:** The authors declare no competing interests.

