## [Peer Review File · Nature Communications]

Reviewers' comments:

Reviewer #1 (Remarks to the Author):

Using a conditional Myrf KO in oligodendrocytes, this manuscript definitively shows that remyelination is not essential for hindlimb locomotor recovery after a mid-thoracic contusive SCI. The experiments are well conceived, data for the most part properly analyzed, and the manuscript itself is extremely well written. These data are important as myelinating cell grafts have been reported by many investigators to improve locomotor function after these injuries. In light of the current findings, the conclusions from those studies will have to be re-interpreted. These data emphasize that the important variables for the extent of recovery after thoracic contusive SCI are the extent of spared white matter and the reorganization of the lumbar locomotor circuitry. While this conclusion has been inferred from previous studies, current data definitively rule out remyelination as a reparative process that impacts on the extent of recovery. These data thus have important implications with respect to rationale design of cellular therapies for SCI. Individual concerns are detailed below.

Lines 61-2: The wrong verb is used. I believe the context would be better conferred in the authors used 'identify' not 'act as' and 'which differentiate into' rather than 'produce'.

Neither the methods nor the results indicate if the Pdgfr-CreERT2:Myrffl/fl mice were heterozygous or homozygous on the two transgenes. This must be provided.

There is inconsistent nomenclature with respect to how the mouse gene and proteins are denoted. The authors are referred to the following website for current guidelines.

<http://www.informatics.jax.org/mgihome/nomen/gene.shtml>

Line 102+: The authors do not use the most appropriate controls in these studies. Ideally, they should use Pdgfr-CreERT2:Myrffl/fl mice + vehicle, rather than Myrffl/fl mice + tamoxifen. While I appreciate the difficulties in breeding double transgenics, double transgenics often show phenotypes that are distinct from the parent floxed strain. At the very least, they must show that there is no difference between these two controls.

Line 408+: To unequivocally count the absolute number of objects in a section, unbiased stereology must be used (e.g. PMID: 28798525, PMID: 25743692, PMID: 25743692). Those methods are not used here. The authors should just why they were not and why the methods that did employ were indeed devoid of 2D counting artifacts.

There are numerous places throughout this manuscript where the authors use quantitative descriptors and show only single histological images. If the authors want to use the quantitative word 'majority', quantification with appropriate statistical analyses must be done. It is hard to believe that Fig 5d would have nearly complete classes of thin, none, and thick myelin respectively. These sections seem somewhat cherry picked. The authors should provide quantitative data across multiple animals.

Many of the figures and figure legends are not consistent in their use of lower and upper case labels. The authors should correct those inconsistencies.

Figure legend 2, line 4: Olig1+ and CC1+ should be Olig1, CC1.

In Fig 7, the schematics of oligodendrocyte myelin and Schwann cell myelin do not appear to accurately reflect the quantitative data in Figs. 2 & 4. They should be better drawn to scale.

Reviewer #2 (Remarks to the Author):

This is a very interesting study that is well-conducted and with conclusions supported by convincing data. It is also a very important study that should, it is to be hoped, put to bed the notion promoted by some that traumatic spinal cord injury will benefit from pro-remyelination interventions. My main issue concerns are less the way the study has been done but rather the way that the rationale is presented. Personally, I think the authors can more critical of the evidence for persistent demyelination in SCI on which much of the claims that remyelination-enhancing therapies will be useful are based. For example, the paper by Totoui and Keirstead (J Comp Neurol) contains an image claiming persistent demyelination that is widely recognised as misleading. I think the bold statement that chronic oligodendrocyte loss and demyelination is a feature of SCI (lines 54-55 page 2) is not one that many would agree with – and the phrasing should perhaps be tempered. It also somewhat contradicts the claim made in line 70-72 on page 3. Surely the issue here is that 1) when demyelination occurs it is generally followed by rather efficient remyelination (the paper by Jeffery and Smith in Brain Pathol 2006 provides convincing evidence that this occurs in clinical disease that supports the strong body of experimental data and should perhaps be cited), and 2) it is such a minor component of the overall pathology of SCI (compared with frank axonal loss for example) that it is unlikely to be a major driver of functional recovery?

Minor points

1. Many in the field prefer the term oligodendrocyte progenitor cell rather than oligodendrocyte precursor cell, since the term precursor implies a lineage restriction that is not a feature of adult OPCs (the authors own recent work on the origin of Schwann cells in the CNS being part of the evidence that these cells are not lineage restricted).
2. Page 10, line 264 – ‘it is’ rather than ‘it’s’
3. The role of astrocytes in constraining the expansion of Schwann cell remyelination in the absence of oligodendrocyte remyelination should probably be given more prominence (see, for example, Monteiro de Castro et al Am J Pathol 2015).

Reviewer #3 (Remarks to the Author):

After traumatic injury to the spinal cord, some axons crossing the injury site are severed or die back post-injury while others survive but become demyelinated post-injury. The question has arisen, therefore, whether preventing this demyelination or encouraging rapid and more extensive remyelination might be a useful therapeutic goal. The field appears divided on this. The current article by Duncan et al. tries to resolve this issue by examining the role of oligodendrocyte (OL) and myelin production in the limited spontaneous locomotor recovery that follows experimental traumatic injury to mouse spinal cord.

The authors prevented production of new OLs and OL-derived myelin by conditional deletion of the transcription factor Myrf in OL precursors (OPCs) using Pdgfra-CreER(T2), an approach that was devised previously to examine the role of new OLs in motor learning. The present study is well-controlled and demonstrates that new OL production is knocked down >90% in the spared ventrolateral white matter, following a contusion injury to the dorsal funiculus. This resulted in a ~50% reduction in remyelination of spared axons at the injury site, almost all of the observed remyelination being attributable to newly-generated Schwann cells. These Schwann cells were generated within 2 weeks post-injury, mainly from Pdgfra-positive precursors (possibly OPCs) since they were labelled by the Rosa-mGFP reporter. This is something that has been observed previously by the Tetzlaff lab following spinal cord injury, and by others in the context of gliotoxin-

induced demyelination. This Schwann-cell-mediated remyelination, rather than OL-mediated remyelination, might be what drives locomotor recovery in the first 2 weeks post-injury, because OL-mediated remyelination occurs subsequent to that. Production of Schwann cells and Schwann cell-derived myelin was not diminished by conditional KO of *Myrf* in the present experiments so this study does not rule out the possibility that Schwann cell remyelination is important in locomotor recovery. This could be examined in future by an analogous approach using e.g. *Post-CreER(T2)*.

An interesting and useful side-shoot of the study is the observation that almost all OL-mediated remyelination is prevented by deletion of *Myrf*, providing strong evidence that OL remyelination is via newly-generated OLs (from OPCs) rather than by elaboration of new myelin sheaths by pre-existing OLs. Perhaps more could be made of this – in the Discussion for example – with reference to previous attempts to address this very issue (e.g. Crawford et al., 2016 *Am J Pathol* 186, 511).

Overall, I thought that this is an excellent study, well-conducted, well thought-through and well written on the whole, although the text could be shortened significantly if required. The main conclusion, that the limited locomotor recovery observed in the first few weeks of a contusion injury does not rely on OL-mediated remyelination seems sound, and could be important by casting doubt on the utility of attempting to improve outcomes by targeting OL-remyelination. It also focusses attention on the possible role of Schwann cells in functional recovery.

Minor points:

1. In Figures 1A and 3A, the lox sites should be re-drawn in the same orientation. In the opposite orientation as shown, cre-recombination would cause flip-flopping of the intervening DNA, not deletion. Conventionally the direction is shown 5' to 3' (L to R).
2. In most figures the labelling of some panels is far too small to be visible at final size.

Response to Editor and Reviewers

We would like to thank the reviewers and editor for finding time in their busy schedules to read our article and provide insightful comments and critiques. These comments have been the basis for our revisions which have greatly strengthened this manuscript.

Reviewer comments are in bold. Author responses are in plain text. *Statements from the manuscript that address reviewers' comments are italicized, and new text in the manuscript is italicized and bolded.*

Reviewers' comments:

Reviewer #1 (Remarks to the Author):

Using a conditional Myrf KO in oligodendrocytes, this manuscript definitively shows that remyelination is not essential for hindlimb locomotor recovery after a mid-thoracic contusive SCI. The experiments are well conceived, data for the most part properly analyzed, and the manuscript itself is extremely well written. These data are important as myelinating cell grafts have been reported by many investigators to improve locomotor function after these injuries. In light of the current findings, the conclusions from those studies will have to be re-interpreted. These data emphasize that the important variables for the extent of recovery after thoracic contusive SCI are the extent of spared white matter and the reorganization of the lumbar locomotor circuitry. While this conclusion has been inferred from previous studies, current data definitively rule out remyelination as a reparative process that impacts on the extent of recovery. These data thus have important implications with respect to rationale design of cellular therapies for SCI. Individual concerns are detailed below.

We thank the reviewer for their encouraging comments regarding the impact of this study. We agree that previous cell transplantation studies, which were thought to confer functional benefits through remyelination, will need to be re-interpreted. We highlight the significance of these data to cellular transplantation studies and more broadly to whether remyelination is a validated clinical target at several points in the manuscript:

Lines 77-78: *'Despite this, myelin regeneration is the mechanistic basis of several ongoing clinical trials and has become an important^{16,19}, yet contentious^{17, 18, 42} therapeutic target.'*

Lines 323-326: *'However, both the time course of oligodendrocyte remyelination, and the unimpaired recovery relative to control mice in the absence of oligodendrocyte remyelination question both the role of remyelination in this model to drive recovery and the viability of this model to test remyelinating therapies. Ultimately, this study raises doubts whether remyelination is a validated target for clinical translation following moderate spinal cord contusion.'*

Lines 61-2: The wrong verb is used. I believe the context would be better conferred in the authors used 'identify' not 'act as' and 'which differentiate into' rather than 'produce'.

We agree with the reviewer and have used the suggested verbs to clarify the sentence. We have changed it from:

Line 60-61: *Platelet-derived growth factor receptor α (PDGFR α), act as oligodendrocyte precursor cells (OPCs) to produce new oligodendrocytes after SCI^{20, 22, 26}*

To

Line 60-62: *Platelet-derived growth factor receptor α (PDGFR α) expression in resident, non-vascular associated cells, identifies these cells as oligodendrocyte progenitor cells (OPCs)^{28, 29}, which differentiate into new oligodendrocytes after SCI^{20, 22, 26}.*

Neither the methods nor the results indicate if the Pdgfr-CreERT2:Myrffl/fl mice were heterozygous or homozygous on the two transgenes. This must be provided.

We agree that this is an important methodological detail that was overlooked in the initial manuscript. We have now added whether the gene was heterozygous or homozygous for each transgene used in both the results and methods section of the manuscript.

Line 98-99 (**bolded** is new text): *'We crossed mice carrying LoxP sites flanking **both copies of exon 8 (homozygous)** of the Myrf gene (Myrf^{fl/fl})...'*

Line 109-110: *'We examined the effectiveness of tamoxifen to induce recombination in the spinal cord of both Myrf^{wt/wt} and Myrf^{fl/fl} mice **heterozygous** for the PDGFR α -CreERT2 and the Rosa26 mGFP (mT/mG) **transgenes** after injury.'*

Line 333-334: *'Mice from the parental Myrf^{fl/fl} line (Jackson Laboratory stock # 010607)³², which express LoxP sites around **both copies** of exon 8 of Myrf (**homozygous**)...'*

Line 340: *'All Myrf **ICKO** mice used were heterozygous for PDGFR α -CreERT2.'*

Line 348-350: *'Myrf **ICKO** and control mice were heterozygous for the mT/mG and PDGFR α -CreERT² transgenes in genetic fate mapping experiments.'*

There is inconsistent nomenclature with respect to how the mouse gene and proteins are denoted. The authors are referred to the following website for current guidelines. <http://www.informatics.jax.org/mgihome/nomen/gene.shtml>

We thank the reviewer for this suggestion and have corrected all cases of nomenclature for both genes and proteins that did not adhere to guidelines in both the text and figures. All gene and

allele names now have their first letter capitalized and the whole name is italicized, while protein symbols have been capitalized in alignment with these conventions.

Line 102+: The authors do not use the most appropriate controls in these studies. Ideally, they should use *Pdgfr-CreERT2:Myr^{fl/fl}* mice + vehicle, rather than *Myr^{fl/fl}* mice + tamoxifen. While I appreciate the difficulties in breeding double transgenics, double transgenics often show phenotypes that are distinct from the parent floxed strain. At the very least, they must show that there is no difference between these two controls.

To address the reviewer's concern, and to ensure PDGFR α transgene insertion alone did not alter recovery after SCI, we conducted an additional experiment in which *Myr^{fl/fl}* mice heterozygous with the PDGFR α -CreERT2 transgene were compared to littermate *Myr^{fl/fl}* mice without the PDGFR α -CreERT2 transgene in the absence of tamoxifen. These mice did not differ in their response to T9/10 thoracic spinal cord injury, indicating there is no inherent difference between these two controls (Supplementary Figure 1 – see below).

Supplementary Fig. 1

Myrf^{fl/fl} mice do not inherently differ in locomotor recovery from moderate thoracic SCI relative to *Myrf^{fl/fl} PDGFR α CreERT2* in the absence of tamoxifen. (a) Measurement of impact force imparted on the spinal cord of *Myrf^{fl/fl}* and *Myrf^{fl/fl} PDGFR α CreERT2* reveals no differences between groups ($df=15$, $t=0.604$, $P=0.555$, Student's *t*-test). (b) A graph of the displacement of the impactor tip into the spinal cord during thoracic contusion indicates no distinction between *Myrf^{fl/fl}* and *Myrf^{fl/fl} PDGFR α CreERT2* mice ($df=15$, $t=0.318$, $P=0.755$, Student's *t*-test). Graphs of open field performance as assessed on the (c) BMS, and (d) BMS subscore indicates no differences in hindlimb recovery between *Myrf^{fl/fl}* and *Myrf^{fl/fl} PDGFR α CreERT2* in the absence of tamoxifen (BMS $F(1,15)=0.004$, $P=0.951$; BMS subscore $F(1,15)=0.371$, $P=0.552$, two-way repeated measures ANOVA).

Line 342-345 added to text: 'All animals, unless stated otherwise, were treated with tamoxifen to ensure no confounding effects of the drug on either recovery from SCI or remyelination efficiency. The insertion of the PDGFR α -CreERT2 transgene does not affect recovery following contusive SCI relative to *Myrf^{fl/fl}* lacking the PDGFR α -CreERT2 in the absence of tamoxifen (Supplementary Fig. 1)'

We recognize that the transgenic mice breeding strategy could have been articulated more clearly in the text. As a result we have added or changed the following lines.

Lines 333-340: *‘Mice from the parental Myrf^{fl/fl} line (Jackson Laboratory stock # 010607)³², which express LoxP sites around both copies of exon 8 of Myrf (homozygous), were crossed with PDGFR α -CreERT2⁴⁴ (Jackson Laboratory stock # 018280) mice. Offspring from the F1 generation with the PDGFR α -CreERT2 transgene and heterozygous for the presence of LoxP sites around Myrf (Myrf^{fl/wt}) were crossed with F1 Myrf^{fl/wt} mice lacking the PDGFR α -CreERT2 transgene. F2 generation Myrf^{fl/fl} PDGFR α -CreERT2 and Myrf^{fl/fl} mice lacking the Cre transgene were then bred to produce sufficient mice for the experiments. This breeding strategy yielded litters in which all mice had both copies of Myrf surrounded by LoxP sites (Myrf^{fl/fl}) with individual mice either without (control mice) or with the PDGFR α -CreERT2 transgene (Myrf ICKO). ’*

This description indicates that all control mice used in the experiments were littermates of Myrf ICKO mice. This breeding strategy ensured that the control mice were on the same genetic background as the knockout mice. It had the added benefit of being ideal for blinding during behavioural experiments, as cages would contain both control and knockout animals.

Additionally, using Myrf^{fl/fl} PDGFR α -CreERT2-negative mice administered tamoxifen as controls for behavioural experiments avoided known confounds that tamoxifen has on recovery. Tamoxifen is a highly biologically active molecule. At lower doses, tamoxifen has been demonstrated by several groups to be neuroprotective after spinal cord injury (Williams et al., 1996; Tian et al., 2009; Guptarak et al., 2014; Colon et al., 2016; de la Torre Valdovinos et al., 2016; Osuna-Carrasco et al., 2016), traumatic brain injury (Franco Rodriguez et al., 2013) and can even enhance remyelination by directly acting on oligodendrocyte progenitor cells (Gonzalez et al., 2016). Higher doses, like those administered for inducible Cre-LoxP experiments, yield increased cellular stress in select neuronal populations (Denk et al., 2015). By administering tamoxifen to all mice used in behavioural analyses, we control for its known biological effect.

Line 408+: To unequivocally count the absolute number of objects in a section, unbiased stereology must be used (e.g. PMID: 28798525, PMID: 25743692, PMID: 25743692). Those methods are not used here. The authors should just why they were not and why the methods that did employ were indeed devoid of 2D counting artifacts.

We appreciate the reviewer highlighting the importance of using proper stereological methodology in counts of absolute objects in a section. We agree that without the use of stereological principles there is an increased probability of counting artifacts.

In this manuscript we did counts of oligodendrocyte density in thick (3D) sections and 2D counts of myelin from thin sections. All of our counts of objects (either cells, or myelin sheaths) used the key principles of stereology to reduce counting artifacts and bias. For example, all 3D counts are conducted using an optical disector to guard against overestimates or duplicate object counts by counting only unique elements. This ensures that there is not a change in estimate due to an alteration in the size, shape or orientation of the cell. The experimental details from the manuscript demonstrating that stereological principles were used are highlighted in further detail

below. Additionally, we discuss the stereology of our counts of relative cell density in 3D space versus the absolute counts of myelin sheaths from 2D sections. We recognize now that elements could have been stated more explicitly and these points are included within the methods section.

A key aspect of stereology is that systematic uniform random sampling (SURS) is employed (Gundersen et al., 1988; Brown, 2017). In all of our 3D cell counts, the entirety of a given section was uniformly sampled, as indicated in the methods, to ensure that objects had an equal chance of being counted (***bolded is added text***).

Line 432-434: *‘We performed systematic **uniform** random sampling within each section⁶⁹ by overlying a grid (individual grid size 103 μm x 108 μm) onto a low magnification preview image of a cross section of spinal cord. One counting square for every 3 x 3 grid area was imaged at 400x magnification **with a randomized start location** (40x objective NA 1.3).’*

Sections were also systematically sampled in the rostral-caudal orientation every 200 μm from lesion epicenter. Our goal in these cell counts was to determine the relative density and capacity of *Myrf*ICKO mice to generate new oligodendrocytes, which primarily occurs in close proximity to the lesion (Tripathi and McTigue, 2007; Hesp et al., 2015). For this reason, sections underwent systematic uniform random sampled in the epicenter and 200 or 400 μm rostral and caudal for counts of oligodendrocytes.

Lines 431-432: *‘For analysis of cell densities, we imaged the epicenter of injury and the next two sections 200 and 400 μm rostral and caudal for each animal for a total of five sections per mouse.’*

Crucially, for all counts of oligodendrocyte cell density from thick sections, optical disectors were used to ensure that only unique objects were counted. The entire Z-stack through a section was taken on a confocal microscope before cells were counted within an optical disector as in (Gundersen et al., 1988). Nuclei that came into focus were only counted, and nuclei at the edge of the optical disector were excluded if they touched two of the adjacent sides and included if they touched either of the two opposing sides. Further details (***bolded italics***) have been added to the methods section.

Line 436-437: *‘Z-stacks were imaged through the entire depth of the 20 μm thick section with 1 μm spacing between optical sections and cells were counted in three dimensional space within a 100 x 100 μm optical disector. **Nuclei that came into focus and were within the optical disector or in contact with right and upper edge were counted to ensure only unique objects were quantified.**’*

Therefore, we combined SURS with stereological counting probes in thick 3D sections to reduce sampling artifacts in these counts. This is now clear within the text.

For 2D counts of total myelin content, myelin was quantified at the lesion epicenter because both the level of demyelination and remyelination is highest. Therefore, at this location myelin dynamics are most likely to impact locomotor recovery. The gold standard for measuring either axon number (Larsen, 1998; Zarei et al., 2016) or myelin thickness (g-ratio) is using 2D cross-sections of axons. 2D counts combined with systematic sampling have been used to determine the number of axons within the tibial and optic nerve (Williams et al., 1996; Larsen, 1998).

We sectioned every 20 μm through our resin blocks to guarantee the area with the most severe pathology was measured to ensure that the lesion epicenter was compared between mice. For these counts, systematic uniform random sampling was again employed as well as a counting frame was used with 2 inclusion and 2 exclusion lines to ensure only unique objects were counted. This approach should greatly reduce the chance that individual myelinated axons were oversampled. Larsen and colleagues estimated that in the tibial nerve employing SURS and ensuring that at least 150-200 axons were counted was sufficient to reduce empirical variance to just 5% (Larsen, 1998). We adapted this protocol for use in the spinal cord, but counted a much larger 1500-2500 axons per animal due to the variability in axon density regionally across the injured cord. The area we counted was equivalent to 1/7 of the area of the spinal cord.

Lines 456-458: *'The entire cross section at epicenter was imaged. A grid with box dimensions of 50 μm x 50 μm was overlaid on top of the merged image. We employed systematic random sampling, counting one in every seven grids.'*

There are numerous places throughout this manuscript where the authors use quantitative descriptors and show only single histological images. If the authors want to use the quantitative word 'majority', quantification with appropriate statistical analyses must be done. It is hard to believe that Fig 5d would have nearly complete classes of thin, none, and thick myelin respectively. These sections seem somewhat cherry picked. The authors should provide quantitative data across multiple animals.

We have now quantified myelin thickness using electron microscopy. We find that on average myelin tends to be thinner in control mice relative to *Myrf*ICKO throughout the spinal cord at 6 WPI (Figure 5e, f). The paucity of examples of thin myelin sheaths in *Myrf*ICKO (g ratio > 0.85), which are normally rarely found in the uninjured mouse spinal cord (James et al., 2011; Ishii et al., 2014), are suggestive of little remyelination (Figure 5e). Combined with increased number of unmyelinated axons > 1 μm in *Myrf*ICKO (8336 \pm 1072 axons relative to 1298 \pm 327 in controls), provides compelling evidence that remyelination is ablated and chronic demyelination is present (Figure 5g). Broadly, these data indicate that images in figure 5d are representative and *Myrf*ICKO rarely have evidence of thinly myelinated fibers and outright demyelination is widely observed at 6 WPI. We thank the reviewer for this suggestion and think

these data have strengthened the argument that *Myrf*ICKO induces chronic demyelination following SCI.

(e) Frequency distribution of g-ratios of myelinated axons indicating a shift towards higher g-ratios (more thinly myelinated axons) in the controls relative to *Myrf*ICKO ($P < 0.001$, Kolmogorov-Smirnov test). (f) Scatter plot comparing g-ratio to axon diameter of axons quantified in the spared white matter of injured animals demonstrating a difference between controls and *Myrf*ICKO ($F = 25$, $DF_n = 1$, $DF_d = 1340$, $P < 0.0001$, linear regression). Dashed box highlights axons that lack myelin (g) Quantification showing more unmyelinated axons larger than 1 μm in the spared white matter of *Myrf*ICKO compared to controls at 6 WPI ($df = 6$, $t = 4.858$, $P = 0.003$, Student's *t*-test). * = $P \leq 0.05$, ** = $P \leq 0.01$ Scale bars = 100 μm (a), 5 μm (b), 1 μm (f).

Many of the figures and figure legends are not consistent in their use of lower and upper case labels. The authors should correct those inconsistencies.

Lower and upper case lettering has now been made consistent in the figure and figure legends throughout the manuscript.

Figure legend 2, line 4: Olig1+ and CC1+ should be Olig1, CC1.

This typographic error has been corrected.

In Fig 7, the schematics of oligodendrocyte myelin and Schwann cell myelin do not appear to accurately reflect the quantitative data in Figs. 2 & 4. They should be better drawn to scale.

We have changed the schematic in Fig. 7b to reflect the actual quantified values in the manuscript for myelin. Likewise, the BMS scores follow the recovery curves of the group means over time. We felt this figure was crucial for highlighting the temporal discordance between remyelination and behavioural recovery.

Reviewer #2 (Remarks to the Author):

This is a very interesting study that is well-conducted and with conclusions supported by convincing data. It is also a very important study that should, it is to be hoped, put to bed the notion promoted by some that traumatic spinal cord injury will benefit from pro-remyelination interventions.

My main issue concerns are less the way the study has been done but rather the way that the rationale is presented. Personally, I think the authors can more critical of the evidence for persistent demyelination in SCI on which much of the claims that remyelination-enhancing therapies will be useful are based. For example, the paper by Totoui and Keirstead (J Comp Neurol) contains an image claiming persistent demyelination that is widely recognised as misleading. I think the bold statement that chronic oligodendrocyte loss and demyelination is a feature of SCI (lines 54-55 page 2) is not one that many would agree with – and the phrasing should perhaps be tempered.

We agree with the reviewer that there is little evidence of persistent demyelination after chronic SCI. We have stated as much in recent reviews (Plemel et al., 2014; Assinck et al., 2017a). We did not intend to imply that chronic demyelination was a feature of SCI, only that oligodendrocyte loss occurred for several weeks after SCI. However, we recognize this phrasing could be construed as such and so we have reworded the sentence to ensure our rationale is clear.

Lines 54-55 changed from: *‘However, chronic oligodendrocyte death⁶ and demyelination of spared axons are characteristic after SCI^{7, 8, 9, 10} and could diminish connectivity of spared circuits’*

To:

‘However, oligodendrocyte death in the weeks after SCI⁶ presumably results in the demyelination of spared axons^{7,8,9,10}, which could diminish the connectivity of spared circuits.’

It also somewhat contradicts the claim made in line 70-72 on page 3. Surely the issue here is that 1) when demyelination occurs it is generally followed by rather efficient remyelination (the paper by Jeffery and Smith in Brain Pathol 2006 provides convincing evidence that this occurs in clinical disease that supports the strong body of experimental data and should perhaps be cited), and 2) it is such a minor component of the overall pathology of SCI (compared with frank axonal loss for example) that it is unlikely to be a major driver of functional recovery?

We have amended lines 54-55 (see previous response) so that we no longer contradict ourselves in lines 70-72. Many studies demonstrate that remyelination is a highly efficient process after SCI, including this one and previous work from our laboratory (Powers et al., 2012; Assinck et

al., 2017b). We have now cited the manuscript mentioned above as it is clear evidence of de/remyelination in a clinically intermediate-sized animal model.

Lines 72-74 *‘Endogenous myelin regeneration is an efficient process after SCI, as indicated by the presence of numerous thinly myelinated axons^{10, 13, 19} shorter internodes^{9, 24, 40} and by fluorescently labeling new myelin in transgenic mice^{22,26}. **Citation added***

We thank the reviewer for highlighting these problems with how the rationale is presented, and think it is now much clearer and more accurate.

Minor points

1. Many in the field prefer the term oligodendrocyte progenitor cell rather than oligodendrocyte precursor cell, since the term precursor implies a lineage restriction that is not a feature of adult OPCs (the authors own recent work on the origin of Schwann cells in the CNS being part of the evidence that these cells are not lineage restricted).

We have changed the term ‘precursor’ to ‘progenitor’ in all instances in the text.

2. Page 10, line 264 – ‘it is’ rather than ‘it’s’

This change has been implemented.

3. The role of astrocytes in constraining the expansion of Schwann cell remyelination in the absence of oligodendrocyte remyelination should probably be given more prominence (see, for example, Monteiro de Castro et al Am J Pathol 2015).

We agree that astrocytes likely have a prominent role in constraining Schwann cell myelination after SCI. We have elaborated on this point in the discussion, and added the citation mentioned. We also pointed out how the *Myrf* ICKO did not alter astrocyte coverage, which could have restricted Schwann cell compensation following oligodendrocyte remyelination failure with *Myrf* ICKO.

Reviewer #3 (Remarks to the Author):

After traumatic injury to the spinal cord, some axons crossing the injury site are severed or die back post-injury while others survive but become demyelinated post-injury. The question has arisen, therefore, whether preventing this demyelination or encouraging rapid and more extensive remyelination might be a useful therapeutic goal. The field appears divided on this. The current article by Duncan et al. tries to resolve this issue by examining the role of oligodendrocyte (OL) and myelin production in the limited spontaneous

locomotor recovery that follows experimental traumatic injury to mouse spinal cord.

The authors prevented production of new OLs and OL-derived myelin by conditional deletion of the transcription factor Myrf in OL precursors (OPCs) using Pdgfra-CreER(T2), an approach that was devised previously to examine the role of new OLs in motor learning. The present study is well-controlled and demonstrates that new OL production is knocked down >90% in the spared ventrolateral white matter, following a contusion injury to the dorsal funiculus. This resulted in a ~50% reduction in remyelination of spared axons at the injury site, almost all of the observed remyelination being attributable to newly-generated Schwann cells. These Schwann cells were generated within 2 weeks post-injury, mainly from Pdgfra-positive precursors (possibly OPCs) since they were labelled by the Rosa-mGFP reporter. This is something that has been observed previously by the Tetzlaff lab following spinal cord injury, and by others in the context of gliotoxin-induced demyelination. This Schwann-cell-mediated remyelination, rather than OL-mediated remyelination, might be what drives locomotor recovery in the first 2 weeks post-injury, because OL-mediated remyelination occurs subsequent to that. Production of Schwann cells and Schwann cell-derived myelin was not diminished by conditional KO of Myrf in the present experiments so this study does not rule out the possibility that Schwann cell remyelination is important in locomotor recovery. This could be examined in future by an analogous approach using e.g. Po-CreER(T2).

We thank the reviewer for the kind words on the construct and importance of this study. We agree that Schwann cell myelination may be a driver of functional recovery and stated this explicitly in the text:

Line 307-309: *‘Importantly, we found Schwann cell myelination, in contrast to oligodendrocyte remyelination, occurs early enough after injury to potentially mediate recovery.’*

An interesting and useful side-shoot of the study is the observation that almost all OL-mediated remyelination is prevented by deletion of Myrf, providing strong evidence that OL remyelination is via newly-generated OLs (from OPCs) rather than by elaboration of new myelin sheaths by pre-existing OLs. Perhaps more could be made of this – in the Discussion for example – with reference to previous attempts to address this very issue (e.g. Crawford et al., 2016 Am J Pathol 186, 511).

We agree with the reviewer that this is an important point and have emphasized at several points in the manuscript. We have also added the citation mentioned above to the discussion and the following lines (*bolded italics*).

Lines 260-263: *‘Oligodendrocyte genesis by resident PDGFR α + OPCs cannot be compensated for by other cell sources like ependymal cells or Schwann cells, even when resident OPC*

differentiation is blocked following SCI. Therefore, PDGFR α + progenitors, have an essential role in the generation of new oligodendrocyte myelin after SCI analogous to their role in chemical demyelination⁵²?

Overall, I thought that this is an excellent study, well-conducted, well thought-through and well written on the whole, although the text could be shortened significantly if required. The main conclusion, that the limited locomotor recovery observed in the first few weeks of a contusion injury does not rely on OL-mediated remyelination seems sound, and could be important by casting doubt on the utility of attempting to improve outcomes by targeting OL-remyelination. It also focusses attention on the possible role of Schwann cells in functional recovery.

Minor points:

1. In Figures 1A and 3A, the lox sites should be re-drawn in the same orientation. In the opposite orientation as shown, cre-recombination would cause flip-flopping of the intervening DNA, not deletion. Conventionally the direction is shown 5' to 3' (L to R).

We apologize for this error, and have corrected this in Figures 1A and 3A.

2. In most figures the labelling of some panels is far too small to be visible at final size.

We have made an effort in all of the figures to increase the font size where possible.

References

- Assinck P, Duncan GJ, Hilton BJ, Plemel JR, Tetzlaff W (2017a) Cell transplantation therapy for spinal cord injury. *Nat Neurosci* 20:637-647. 10.1038/nn.4541
- Assinck P, Duncan GJ, Plemel JR, Lee MJ, Stratton JA, Manesh SB, Liu J, Ramer LM, Kang SH, Bergles DE, Biernaskie J, Tetzlaff W (2017b) Myelinogenic Plasticity of Oligodendrocyte Precursor Cells following Spinal Cord Contusion Injury. *J Neurosci* 37:8635-8654. 10.1523/JNEUROSCI.2409-16.2017
- Brown DL (2017) Bias in image analysis and its solution: unbiased stereology. *Journal of toxicologic pathology* 30:183-191. 10.1293/tox.2017-0013
- Colon JM, Torrado AI, Cajigas A, Santiago JM, Salgado IK, Arroyo Y, Miranda JD (2016) Tamoxifen Administration Immediately or 24 Hours after Spinal Cord Injury Improves Locomotor Recovery and Reduces Secondary Damage in Female Rats. *Journal of neurotrauma* 33:1696-1708. 10.1089/neu.2015.4111
- de la Torre Valdovinos B, Duenas Jimenez JM, Estrada IJ, Banuelos Pineda J, Franco Rodriguez NE, Lopez Ruiz JR, Osuna Carrasco LP, Candanedo Arellano A, Duenas Jimenez SH (2016) Tamoxifen Promotes Axonal Preservation and Gait Locomotion Recovery after Spinal Cord Injury in Cats. *Journal of veterinary medicine* 2016:9561968. 10.1155/2016/9561968

- Denk F, Ramer LM, Erskine EL, Nassar MA, Bogdanov Y, Signore M, Wood JN, McMahon SB, Ramer MS (2015) Tamoxifen induces cellular stress in the nervous system by inhibiting cholesterol synthesis. *Acta neuropathologica communications* 3:74. 10.1186/s40478-015-0255-6
- Franco Rodriguez NE, Duenas Jimenez JM, De la Torre Valdovinos B, Lopez Ruiz JR, Hernandez Hernandez L, Duenas Jimenez SH (2013) Tamoxifen favoured the rat sensorial cortex regeneration after a penetrating brain injury. *Brain Res Bull* 98:64-75. 10.1016/j.brainresbull.2013.07.007
- Gonzalez GA, Hofer MP, Syed YA, Amaral AI, Rundle J, Rahman S, Zhao C, Kotter MR (2016) Tamoxifen accelerates the repair of demyelinated lesions in the central nervous system. *Sci Rep* 6:31599. 10.1038/srep31599
- Gundersen HJ, Bagger P, Bendtsen TF, Evans SM, Korbo L, Marcussen N, Moller A, Nielsen K, Nyengaard JR, Pakkenberg B, et al. (1988) The new stereological tools: disector, fractionator, nucleator and point sampled intercepts and their use in pathological research and diagnosis. *APMIS : acta pathologica, microbiologica, et immunologica Scandinavica* 96:857-881.
- Guptarak J, Wiktorowicz JE, Sadygov RG, Zivadinovic D, Paulucci-Holthauzen AA, Vergara L, Nesic O (2014) The cancer drug tamoxifen: a potential therapeutic treatment for spinal cord injury. *Journal of neurotrauma* 31:268-283. 10.1089/neu.2013.3108
- Hesp ZC, Goldstein EA, Miranda CJ, Kaspar BK, McTigue DM (2015) Chronic oligodendrogenesis and remyelination after spinal cord injury in mice and rats. *J Neurosci* 35:1274-1290. 10.1523/JNEUROSCI.2568-14.2015
- Ishii A, Furusho M, Dupree JL, Bansal R (2014) Role of ERK1/2 MAPK signaling in the maintenance of myelin and axonal integrity in the adult CNS. *J Neurosci* 34:16031-16045. 10.1523/JNEUROSCI.3360-14.2014
- James ND, Bartus K, Grist J, Bennett DL, McMahon SB, Bradbury EJ (2011) Conduction failure following spinal cord injury: functional and anatomical changes from acute to chronic stages. *J Neurosci* 31:18543-18555. 10.1523/JNEUROSCI.4306-11.2011
- Kang SH, Fukaya M, Yang JK, Rothstein JD, Bergles DE (2010) NG2+ CNS glial progenitors remain committed to the oligodendrocyte lineage in postnatal life and following neurodegeneration. *Neuron* 68:668-681. 10.1016/j.neuron.2010.09.009
- Larsen JO (1998) Stereology of nerve cross sections. *Journal of neuroscience methods* 85:107-118.
- Osuna-Carrasco LP, Lopez-Ruiz JR, Mendizabal-Ruiz EG, De la Torre-Valdovinos B, Banuelos-Pineda J, Jimenez-Estrada I, Duenas-Jimenez SH (2016) Quantitative analysis of hindlimbs locomotion kinematics in spinalized rats treated with Tamoxifen plus treadmill exercise. *Neuroscience* 333:151-161. 10.1016/j.neuroscience.2016.07.023
- Plemel JR, Keough MB, Duncan GJ, Sparling JS, Yong VW, Stys PK, Tetzlaff W (2014) Remyelination after spinal cord injury: is it a target for repair? *Prog Neurobiol* 117:54-72. 10.1016/j.pneurobio.2014.02.006
- Powers BE, Lasiene J, Plemel JR, Shupe L, Perlmutter SI, Tetzlaff W, Horner PJ (2012) Axonal thinning and extensive remyelination without chronic demyelination in spinal injured rats. *J Neurosci* 32:5120-5125. 10.1523/JNEUROSCI.0002-12.2012
- Tian DS, Liu JL, Xie MJ, Zhan Y, Qu WS, Yu ZY, Tang ZP, Pan DJ, Wang W (2009) Tamoxifen attenuates inflammatory-mediated damage and improves functional outcome after spinal cord injury in rats. *Journal of neurochemistry* 109:1658-1667. 10.1111/j.1471-4159.2009.06077.x
- Tripathi R, McTigue DM (2007) Prominent oligodendrocyte genesis along the border of spinal contusion lesions. *Glia* 55:698-711. 10.1002/Glia.20491

THE UNIVERSITY
OF BRITISH COLUMBIA

Wolfram Tetzlaff, M.D. PhD.

Professor and ICORD Director

ICORD, Blusson Spinal Cord Centre,
818 W 10th Ave. Vancouver, BC Canada V5Z 1M9.

Email: tetzlaff@icord.org

Williams RW, Strom RC, Rice DS, Goldowitz D (1996) Genetic and environmental control of variation in retinal ganglion cell number in mice. *J Neurosci* 16:7193-7205.

Zarei K, Scheetz TE, Christopher M, Miller K, Hedberg-Buenz A, Tandon A, Anderson MG, Fingert JH, Abramoff MD (2016) Automated Axon Counting in Rodent Optic Nerve Sections with AxonJ. *Sci Rep* 6:26559. 10.1038/srep26559

REVIEWERS' COMMENTS:

Reviewer #1 (Remarks to the Author):

The authors have satisfactorily addressed all of my previous concerns. It is an outstanding contribution to the field.

Reviewer #2 (Remarks to the Author):

The authors have attended to all my initial concerns and I have no further issues with this interesting and important paper.

Reviewer #3 (Remarks to the Author):

My only comment after reading the other reviews and the authors' responses is that the Schwann cell remyelination observed in this study has been down-played substantially, possibly because the conclusion of the study can only be that OL-mediated remyelination is not a driver of functional recovery, not that remyelination per se is unimportant. This important point has been missed by reviewer 1 who effuses that "current data definitively rule out remyelination as a reparative process that impacts on the extent of recovery". The title and abstract refer to "oligodendrocyte remyelination", but Schwann cell remyelination is not mentioned up-front and the abstract concludes "remyelination is not required for spontaneous recovery of stepping". This is potentially misleading and I suggest that the potential role of Schwann cell remyelination is made clearly and explicitly in the abstract and discussion, not just in lines 307-309. Ultimately, the role of Schwann cell remyelination will need to be specifically tested using other Cre lines.

Response to Reviewers

Reviewer comments are in bold. Author responses are in plain text. *Statements from the manuscript that address reviewers' comments are italicized, and new text in the manuscript is italicized and bolded.*

Reviewers' comments:

Reviewer #1 (Remarks to the Author): The authors have satisfactorily addressed all of my previous concerns. It is an outstanding contribution to the field.

Reviewer #2 (Remarks to the Author):

The authors have attended to all my initial concerns and I have no further issues with this interesting and important paper.

We thank reviewer 1 and 2 for the enthusiastic comments regarding the importance and quality of the data within this manuscript.

Reviewer #3 (Remarks to the Author):

My only comment after reading the other reviews and the authors' responses is that the Schwann cell remyelination observed in this study has been down-played substantially, possibly because the conclusion of the study can only be that OL-mediated remyelination is not a driver of functional recovery, not that remyelination per se is unimportant. This important point has been missed by reviewer 1 who effuses that "current data definitively rule out remyelination as a reparative process that impacts on the extent of recovery". The title and abstract refer to "oligodendrocyte remyelination", but Schwann cell remyelination is not mentioned up-front and the abstract concludes "remyelination is not required for spontaneous recovery of stepping". This is potentially misleading and I suggest that the potential role of Schwann cell remyelination is made clearly and explicitly in the abstract and discussion, not just in lines 307-309. Ultimately, the role of Schwann cell remyelination will need to be specifically tested using other Cre lines.

We agree with the reviewer that it is important to distinguish for the reader that this paper only determines the role of oligodendrocyte remyelination in locomotor recovery, not Schwann cell remyelination. We have now ensured that every statement on our results concerning the role of remyelination in locomotor recovery is now prefaced with 'oligodendrocyte.'

For example (***bolded italics is new text***):

Abstract lines 46-48: *‘Collectively, these data demonstrate that MYRF expression in PDGFR α -positive cell derived oligodendrocytes is indispensable for myelin regeneration following contusive SCI but that **oligodendrocyte** remyelination is not required for spontaneous recovery of stepping.’*

Introduction lines 80-81: *‘To ascertain the role of **oligodendrocyte** myelin regeneration in locomotor recovery, we used transgenic mice which permit the selective ablation of oligodendrocyte remyelination.’*

Introduction lines 84-86 (sentence added regarding the impact of Myrf deletion on Schwann cell myelination): *‘**Schwann cell myelination is not altered by Myrf deletion from PDGFR α + cells, nor does the extent of Schwann cell myelination increase to compensate for a failure of oligodendrocyte remyelination.**’*

Introduction lines 90-92: *‘These data indicate that while spontaneous **oligodendrocyte** remyelination is extensive following SCI, it is not associated with improvements in hindlimb motor function during spontaneous recovery in this model.’*

Results lines 96-99: *‘Genetic fate mapping reveals extensive remyelination by resident OPCs differentiating into new oligodendrocytes in response to SCI²⁶, however the extent to which **oligodendrocyte** remyelination contributes to spontaneous motor improvements is unknown.’*

Results lines 100-101: *‘This would enable an assessment of the role of endogenous **oligodendrocyte** remyelination in functional improvements’*

Results lines 223-225: *Thus, Myrf ICKO mice provide the necessary contrast to understand the contribution of **oligodendrocyte** remyelination to locomotor recovery.*

Results lines 248-251: *‘Importantly, when the rate of **oligodendrocyte** remyelination is compared to locomotor improvements, we find the majority of hindlimb recovery following moderate thoracic SCI occurs during the first two weeks, when little oligodendrocyte remyelination is present in mice (Fig. 3g relative to Fig. 6a, and summarized in Fig. 7a, b).’*

Discussion lines 332-335: *‘However, both the time course of **oligodendrocyte** remyelination, and the unimpaired recovery relative to control mice in the absence of oligodendrocyte remyelination question both the role of **oligodendrocyte** remyelination in this model to drive recovery and the viability of this model to test remyelinating therapies.’*

With these changes we have now made it clear throughout the text that this manuscript only provides a functional assessment of oligodendrocyte remyelination, and not the contribution of Schwann cell myelination to recovery. In addition, we have provided a full two paragraphs within the discussion on the role of Schwann cell myelination in recovery and the relationship between the extent of Schwann cell and oligodendrocyte remyelination in the injured spinal cord.

Lines 309-326:

‘Schwann cell myelination was observed within the first two weeks following SCI when the majority of locomotor recovery occurs. Schwann cell myelin within the CNS is sufficient to improve conductance following CNS demyelination⁶⁴ and transplantation of Schwann cells into the injured spinal cord has been reported to confer functional benefits⁴². Consistent with the possibility of Schwann cells potentially driving a portion of recovery is a recent study demonstrating that the inducible knockout of neuregulin-1, which prevents Schwann cell myelination following moderate thoracic SCI, is correlated with diminished functional locomotor recovery²⁷. Importantly, we found Schwann cell myelination, in contrast to oligodendrocyte remyelination, occurs early enough after injury to potentially mediate recovery. However, determining the role of Schwann cell myelination during recovery following SCI still requires future cell-specific knockout experiments.

Interestingly, the ablation of oligodendrocyte remyelination did not induce a compensatory increase in Schwann cell myelination, which was still primarily confined to the dorsal column. This raises the intriguing possibility, that the injury environment leaves different CNS axon populations selectively permissible to either Schwann cell or oligodendrocyte remyelination following SCI. The size of the axon²⁴, and the proximity to peripheral roots⁶⁵ may be factors contributing to Schwann cell generation from OPCs, but astrocytes seem to have the prominent role in regulating the level of Schwann cell myelination^{66, 67}. Schwann cell myelination is confined to areas depleted of astrocytes after SCI⁶⁶, and STAT3-mediated reactive astrogliosis restricts Schwann cell myelination following demyelination⁶⁷. The knockout of Myrf from OPCs does not alter astrocyte coverage in the injured spinal cord, which may have restricted a compensatory increase in Schwann cell myelination.

Due to strict space limitations in the abstract, we were not able to comment on the hypothetical role of Schwann cell myelination in locomotor recovery (about which our paper has no direct data), but have clarified that we only assessed the role of oligodendrocyte remyelination in recovery. We fully agree with the author that the role of Schwann cell myelin in recovery is intriguing given its temporal concordance with recovery and requires carefully controlled gene knockout experiments. We have written:

Discussion lines 314-318: *‘Importantly, we found Schwann cell myelination, in contrast to oligodendrocyte remyelination, occurs early enough after injury to potentially mediate locomotor recovery. However, determining the role of Schwann cell myelination during recovery following SCI still requires future cell-specific knockout experiments.’*